# LOCAL-PROMPT: EXTENSIBLE LOCAL PROMPTS FOR FEW-SHOT OUT-OF-DISTRIBUTION DETECTION

**Fanhu Zeng**[1,2]**, Zhen Cheng**[1,2]**, Fei Zhu**[3]**, Hongxin Wei**[4]**, Xu-Yao Zhang**[1,2*]

[1]State Key Laboratory of Multimodal Artificial Intelligence Systems, Institute of Automation, Chinese Academy of Sciences
[2]School of Artificial Intelligence, University of Chinese Academy of Sciences
[3]Centre for Artificial Intelligence and Robotics, HKISI-CAS
[4]Department of Statistics and Data Science, Southern University of Science and Technology
{zengfanhu2022, chengzhen2019, zhufei2018}@ia.ac.cn
xyz@nlpr.ia.ac.cn, weihx@sustech.edu.cn

## ABSTRACT

Out-of-Distribution (OOD) detection, aiming to distinguish outliers from known categories, has gained prominence in practical scenarios. Recently, the advent of vision-language models (VLM) has heightened interest in enhancing OOD detection for VLM through few-shot tuning. However, existing methods mainly focus on optimizing global prompts, ignoring refined utilization of local information with regard to outliers. Motivated by this, we freeze global prompts and introduce **Local-Prompt**, a novel coarse-to-fine tuning paradigm to emphasize regional enhancement with local prompts. Our method comprises two integral components: global prompt guided negative augmentation and local prompt enhanced regional regularization. The former utilizes frozen, coarse global prompts as guiding cues to incorporate negative augmentation, thereby leveraging local outlier knowledge. The latter employs trainable local prompts and a regional regularization to capture local information effectively, aiding in outlier identification. We also propose regional-related metric to empower the enrichment of OOD detection. Moreover, since our approach explores enhancing local prompts only, it can be seamlessly integrated with trained global prompts during inference to boost the performance. Comprehensive experiments demonstrate the effectiveness and potential of our method. Notably, our method reduces average FPR95 by 5.17% against state-of-the-art method in 4-shot tuning on challenging ImageNet-1k dataset, even outperforming 16-shot results of previous methods. Code is released at https://github.com/AuroraZengfh/Local-Prompt.

## 1 INTRODUCTION

Out-of-distribution (OOD) detection (Liu et al., 2020; Cheng et al., 2023b; Jaeger et al., 2022; Tao et al., 2022) aims to distinguish outliers, *i.e.*, samples that do not belong to known in-distribution (ID) classes. It is crucial for industries that require a high level of safety, such as face recognition (Lopez-Lopez et al., 2022) and autonomous driving (Filos et al., 2020). Most previous methods in the field of OOD detection handle the problem using single-modal methods and concentrate on post-hoc processing (Hendrycks & Gimpel, 2016; Wang et al., 2022) or leveraging outliers (Hendrycks et al., 2018; Jiang et al., 2024). However, they suffer from computational inefficiency or consuming data collection. Recently, as the emergence of vision-language models (Radford et al., 2021; Li et al., 2021) shows promising results in multi-modal tasks including image caption (Li et al., 2022a), video understanding (Xu et al., 2021), and so on, it is promising to exploit textual representations to improve performance of OOD detection given a vision-language model (Radford et al., 2021) as a prior model. Considering that zero-shot undergoes a domain gap between upstream and target distribution while full-tuning may pose threats to the learned representation, it is meaningful to explore few-shot learning for vision-language model in OOD detection, in which the detector is prohibited from real outliers and only has access to several ID images.

---

*Corresponding author.

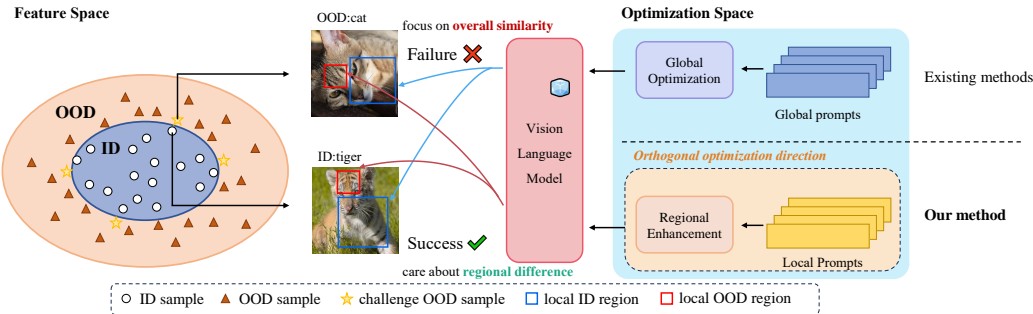

Figure 1: Comparison of prompt learning for OOD detection task. Prompts with global optimization may fail in challenging OOD samples as they are overall similar to ID samples and only have subtle regional differences. For example, cat and tiger are generally similar (blue boxes) and only differ in forehead (red box). Our approach with local outlier knowledge cares about region difference and tackles the issue to some extent.

The most challenging scene for OOD detection is that one hard OOD sample is similar to a known class on the whole and only has subtle differences locally, which naturally requires the detector to identify outliers through local outlier regions. However, existing research falls short of refining OOD task via rich local information when subtle OOD samples are exhibited in certain regions, as is shown in Fig. 1. Some methods merely focus on utilizing global features only (Ming et al., 2022) (blue boxes in Fig. 1), which ignores local features (red boxes in Fig. 1) and inevitably brings about coarse description. Others use the same prompts to match both global and local image features (Miyai et al., 2023b;a), so the gap between them may lead to inaccurate local outlier identification. Conseqently, it is straightforward that enhancing regional information to empower the model with *local outlier knowledge* could be significant to OOD detection.

Motivated by the above observations, we focus on how to explicitly exploit regional-related knowledge, *i.e.*, the learned knowledge is determined by the position of regions, about not only known classes but also unknown samples, which has not been explored before. To tackle the issue, we propose **Local-Prompt**, which concentrates on enhancing local features solely and introduces extensible local prompts for fine local information utilization. Concretely, we decompose global and local prompts to exploit global and local features, and term prompts that interact with global/local features as global/local prompts. Then we propose two integral components: global prompt guided negative augmentation and local prompt enhanced regional regularization. Global prompt is frozen to provide guidance for generating negative augmentation, which could facilitate the exploitation of local outlier knowledge. Fine local prompt is designed for regional regularization to refine regional-related local prompts, which captures the fine local information to better distinguish OOD inputs. Moreover, we also propose corresponding regional-related evaluation metrics to leverage the power of local prompts for OOD detection.

Another advantage of our approach is that it orthogonally explores the benefits of local prompts shown in Fig. 1 and uses hand-crafted global prompts, *i.e.*, a photo of a {class}. Hence, it *is extensible to all existing global prompt designing methods* (Miyai et al., 2023a; Wang et al., 2023). Therefore, we can extend it by seamlessly replacing coarse global prompts with trained ones during inference to improve the performance.

Comprehensive experiments demonstrate the effectiveness of the proposed method. Specifically, our results outperform the state-of-the-art methods in both OOD detection (5.17% reduction on FPR95) and ID accuracy evaluation (1.02%) under 4-shot tuning by a large margin, exhibiting the superiority and the potential of enhancing local prompts for OOD detection and ID classification ability. We additionally carry out various analysis on and all quantitative outcomes and qualitative visualization validate effectiveness. We summarize our contributions as follows:

- We focus on enhancing local information with local prompts and propose a coarse-to-fine few-shot tuning paradigm. Specifically, we propose effective negative augmentation and regional regularization to learn local prompts with local outlier knowledge.

- We propose regional-related OOD score and ID classification metrics, which benefit from enhanced local prompts in OOD detection. Moreover, our method is orthogonal to global

prompt optimization methods and is extensible to get notable performance when integrated with trained local global prompts during inference.

- We conduct comprehensive experiments and achieve competitive results on various OOD detection datasets, demonstrating the superiority of our method. Notably, our 4-shot results on challenging ImageNet-1k even outperform 16-shot results of previous methods.

## 2 RELATED WORK

### 2.1 OUT-OF-DISTRIBUTION DETECTION WITH VISION-LANGUAGE MODEL

Out-of-distribution detection, as a crucial part of open environment (Zeng et al., 2024; Zhu et al., 2023b; Cheng et al., 2023b), has been widely studied in the past decades (Hendrycks et al., 2018; Cheng et al., 2023a; Shu et al., 2023; Zhu et al., 2023a; Liu et al., 2024). With the emergence of CLIP (Radford et al., 2021), which learns outstanding visual and textual representations pre-trained on large-scale image-text pairs and displays excellent ability in various downstream tasks, much attention has been paid to transferring vision-language model to OOD detection task (Fort et al., 2021; Ming et al., 2022; Esmaeilpour et al., 2022; Wang et al., 2023; Miyai et al., 2023a; Park et al., 2023; Bai et al., 2024). For instance, MCM (Ming et al., 2022) adopts the visual and textual representations extracted from CLIP encoders and applies softmax with temperature to better separate ID and OOD data. ZOC (Esmaeilpour et al., 2022) generates pseudo unseen class labels with additional modules and defines a novel confidence score accordingly. Clipn (Wang et al., 2023) enhances the discrimination of OOD samples with extra encoder and data (Sharma et al., 2018).

### 2.2 MULTI-MODAL FEW-SHOT PROMPT LEARNING

Since vision-language model like CLIP (Radford et al., 2021) achieves superior performance in numerous image-text tasks, many attempts have been made to explore few-shot tuning to leverage the power of large pre-trained models. There are mainly two lines of approaches, namely Adapter Learning (Zhang et al., 2021; Gao et al., 2023; Guo et al., 2023; Udandarao et al., 2023; Zhu et al., 2023c) and Prompt Learning (Zhou et al., 2022b; Bahng et al., 2022). Both of them froze the pre-trained encoders when fine-tuning a few additional parameters.

Prompt learning promotes the performance of vision-language model by enhancing the prompts in both visual (Jia et al., 2022; Bahng et al., 2022) and textual (Zhou et al., 2022b;a) aspects. A common practice is setting textual prompts learnable and forwarding them together with visual inputs. CoOp (Zhou et al., 2022b) replaces the hand-crafted prompts with contextual learnable embeddings and optimizes them during few-shot training. In our approach, we freeze global prompts and fine-tune local prompts individually.

### 2.3 GLOBAL AND LOCAL INFORMATION

In Vision Transformer (Dosovitskiy et al., 2020; Zeng & Yu), global tokens are specifically designed for image classification task (Touvron et al., 2021; He et al., 2022). They represent overall characteristics of images. Local tokens are utilized for dense prediction tasks including segmentation (Cheng et al., 2022; Strudel et al., 2021), object detection (Chen et al., 2022; Meng et al., 2021), and so on. The utilization of local features has been explored as well, such as pyramid features (Wang et al., 2021) and window attention (Liu et al., 2021; Li et al., 2022b; Liu et al., 2022).

In the field of OOD detection, global features are primarily employed and numerous studies have been conducted concerning local information (Miyai et al., 2023b;a; Zhang et al., 2023). GL-MCM (Miyai et al., 2023b) takes all tokens into consideration and uses the sum of global and local OOD scores to measure the confidence. LoCoOp (Miyai et al., 2023a) keeps global ID textual embeddings away from the interference of ID-irrelevant regions. However, all existing approaches take same prompts for all features. By contrast, our method directly enhances OOD detection with ID-related areas and refines local prompts to leverage local outlier knowledge.

## 3 PRELIMINARY

### 3.1 PROBLEM DEFINITION

**Few-shot out-of-distribution detection.** Formally, out-of-distribution detection can be viewed as a binary classification that the detector has to identify whether the input image is from ID or OOD space. Labeled training data is composed of $\mathcal{D}_{\text{train}}^{\text{in}} = \{(\boldsymbol{x}_i, y_i)\}_{i=1}^n$, where $\boldsymbol{x}_i$ is sampled *i.i.d.* from joint data distribution space $\mathcal{P}_{\mathcal{X} \mathcal{Y}^{\text{in}}}$ and $y_i \in \mathcal{Y}^{\text{in}}$. $\mathcal{Y}^{\text{in}} = \{1, \cdots, C\}$ is the label space of ID data and $C$ is the number of classes. We transfer the definition of few-shot learning in classification to OOD detection to fine-tune given vision-language model instead of training from scratch. Under few-shot setting, a small proportion of images in each class (*e.g.*, 1, 4, or 16 images) are extracted for training. $\mathcal{Y}^{\text{out}}$ is the label space of OOD data and there is no overlap between ID and OOD label, *i.e.*, $\mathcal{Y}^{\text{in}} \cap \mathcal{Y}^{\text{out}} = \emptyset$. Typically, the model is unable to obtain any OOD data during training process.

The test dataset is a mixture of $\mathcal{D}_{\text{test}}^{\text{in}}$ and $\mathcal{D}_{\text{test}}^{\text{out}}$. When testing, OOD performance is evaluated by discriminating which distribution each sample comes from. A discriminant function is calculated to identify ID and OOD samples:

$$D(\boldsymbol{x}) = \begin{cases} \text{ID}, & S(\boldsymbol{x}) \geq \gamma \\ \text{OOD}, & S(\boldsymbol{x}) < \gamma \end{cases}, \tag{1}$$

where $S(\boldsymbol{x})$ is the confidence score function measuring the uncertainty and $\gamma$ is the threshold.

**Prompt learning with CLIP.** CLIP (Radford et al., 2021) is compromised of image encoder $E_{\text{I}}$ and text encoder $E_{\text{T}}$. Given an image $I$ and a text $T$ describing the corresponding image, they are extracted into feature embeddings, respectively. In prompt learning, textual prompts can be hand-crafted templates (Radford et al., 2021), *e.g.*, "a photo of {class}" or learnable context words (Zhou et al., 2022a), *e.g.*, $\boldsymbol{t}_c = [\boldsymbol{v}_1, \cdots, \boldsymbol{v}_L; \boldsymbol{v}_c]$, where $\boldsymbol{v}_i, (i = 1, \cdots, L)$ is the learnable embedding, and $L$ is the length of context words. $\boldsymbol{v}_c$ ($c \in \{1, \cdots, C\}$) is the embedding of class name. Features of prompts are obtained from text encoder $E_{\text{T}}(\boldsymbol{t}_c) : \mathbb{R}^{(L+1) \times d} \to \mathbb{R}^d$. For simple notation, we use $\boldsymbol{t}_c / \boldsymbol{t} / \hat{\boldsymbol{t}}$ described below to represent features of prompts as text encoder is not the focus of the paper.

Image is first split into several patches with an additional class token embedding $\boldsymbol{x} \in \mathbb{R}^{(N+1) \times d}$. $N$ and $d$ denote the number of local tokens and hidden dimension, respectively. Both global and local visual features are then extracted from image encoder $[\boldsymbol{z}^{\text{g}}, \boldsymbol{z}^{\text{l}}] = E_{\text{I}}(\boldsymbol{x}) : \mathbb{R}^{(N+1) \times d} \to \mathbb{R}^{(N+1) \times d}$, where global features $\boldsymbol{z}^{\text{g}}$ and local features $\boldsymbol{z}^{\text{l}}$ represents overall features and patch-wise regional features, respectively.

In this paper, we denote global prompts and local prompts to hand-crafted/learnable prompts that interact with overall local features and fine local features, respectively.

**OOD detection with CLIP.** The MCM score (Ming et al., 2022) is defined as the maximum of similarity after $\text{softmax}$ with temperature:

$$S_{\text{MCM}}(\boldsymbol{x}) = \max_i \frac{\exp(\text{sim}(\boldsymbol{z}^{\text{g}}, \boldsymbol{t}_i)/T)}{\sum_{j=1}^C \exp(\text{sim}(\boldsymbol{z}^{\text{g}}, \boldsymbol{t}_j)/T)}, \tag{2}$$

and GL-MCM score (Miyai et al., 2023b) considers local information. It applies softmax for local features as well and the maximum is selected as local MCM score. Final OOD score is the sum of MCM score and local MCM score:

$$S_{\text{GL-MCM}}(\boldsymbol{x}) = S_{\text{MCM}}(\boldsymbol{x}) + \max_{i,h} \frac{\exp(\text{sim}(\boldsymbol{z}_h^{\text{l}}, \boldsymbol{t}_i)/T)}{\sum_{j=1}^C \exp(\text{sim}(\boldsymbol{z}_h^{\text{l}}, \boldsymbol{t}_j)/T)}, \tag{3}$$

where $\boldsymbol{z}^{\text{g}}$ represents global feature and $\boldsymbol{z}_h^{\text{l}}$ ($h \in \{1, \cdots, N\}$) represents all extracted local features.

## 4 METHODOLOGY

We propose Local-Prompt, a coarse-to-fine paradigm to strengthen OOD detection with local outlier knowledge. We first present negative augmentation that leverages local and unknown outlier information. Then, we describe local prompt enhancement along with their training regularization and a regional OOD score for better regional information utilization. Finally, we extend our method to trained global prompts to boost the performance. Detailed structures are shown in Fig. 2.

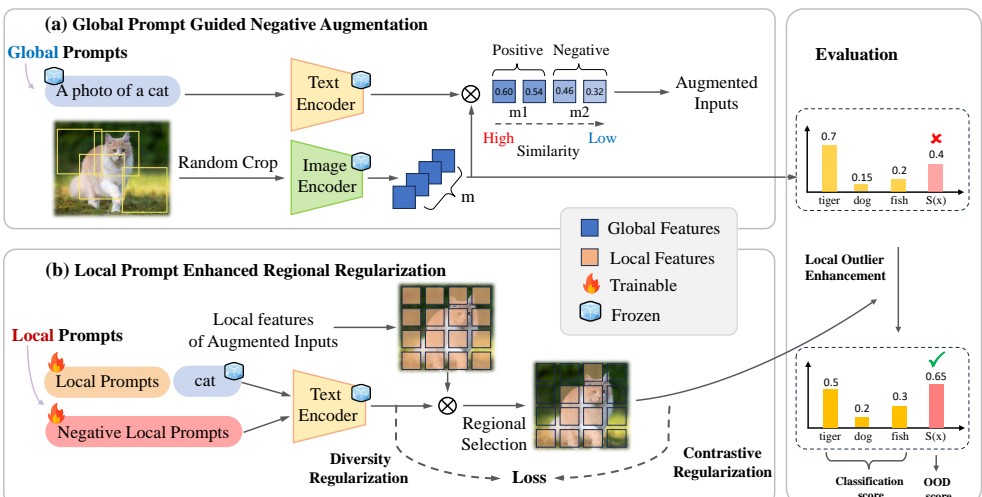

Figure 2: Detailed structure of the proposed Local-Prompt. Our method consists of global prompt guided negative augmentation and local prompt enhanced regional regularization. We froze global prompts to select regional augmented samples and enhance local prompts to learn regional-related representation that helps improve both ID accuracy and OOD detection.

## 4.1 GLOBAL PROMPT GUIDED NEGATIVE AUGMENTATION

As it is costly and not always effective to rely solely on real outliers, one crucial problem for OOD detection is how to leverage imaginary local outlier knowledge to the detector. We propose to synthesize hard negative images to force the model to learn features with strong relation only to simulate outlier situations while avoiding the requirement of real OOD samples. To this end, we apply a simple and straightforward random crop augmentation to empower the detector with unknown (*i.e.*, OOD) information. Specifically, we randomly crop image input $m$ times and use the hand-crafted template with corresponding class name, *i.e.*, $t_c =$ "a photo of {class}", $c = \{1, \cdots, C\}$ as text inputs to calculate the image-text similarity with global features from image encoder. We then select $m_1/m_2$ images with the largest/least similarity, respectively. Images with the largest similarity can be seen as positive samples and those with the least similarity serve as hard negative samples. Notably, random crop is helpful to leverage local information in that local prompts can learn regional-related outlier information through space translation by means of random crop. Effectiveness of random crop is verified in Sec. 5.2.

Global prompts can be viewed as coarse guidance for negative augmentation standing for overall representation and are frozen in our framework. Motivated by previous observations(Ming et al., 2022; Miyai et al., 2023b), we simply use the basic global prompts during training and select augmented images as they are fairly good for extracting global features. Concretely, the global prompts are used to (1) guide the negative augmentation selection, which has been described above; (2) guide the global OOD score calculation for evaluation, as shown in Sec. 4.3.

It is worth emphasizing that our approach is orthogonal to all existing global prompt optimization strategies, *i.e.*, global prompts are built without tuning in our structure (Miyai et al., 2023a), so during inference, it is expected that extending the hand-crafted global prompts with carefully designed ones can promote the performance. Concretely, we replace hand-crafted global prompts with trained ones that have the same shape. Note that our main purpose is to decompose global and local prompts and showcase the effectiveness of local outlier enhancement for OOD detection. Therefore, we do not specifically select the template as it is not the focus of the paper.

## 4.2 LOCAL PROMPT ENHANCED REGIONAL REGULARIZATION

Once augmented inputs are decided, corresponding local features are then used to optimize local prompts to identify ID and OOD samples from the perspective of regional enhancement. We aim at utilizing the local features extracted from image encoder in a fine way to (1) learn regional-related prompts that better characterize local similarity for both ID and OOD regions; (2) enhance detection ability with refined local features. Specifically, we design local prompts $t$ with learnable context

words to represent local textual information for each known class. Moreover, on account of the existence of OOD regions, we build a few local negative prompts $\hat{t}$ to handle possible local outliers.

Contrary to previous global prompt optimization methods, local prompts that work on local features are expected to learn regional-related textual representations and be aware of outliers during training. To this end, we enhance local prompts through regional regularization, which consists of contrastive regularization and diversity regularization, to better distinguish between ID and OOD samples.

**Contrastive regularization.** We propose regional contrastive regularization to enhance local textual prompts for better local information utilization. It is composed of local loss and local negative loss. Given a positive sample of training image, local loss is employed to learn regional information of known classes:

$$\mathcal{L}_{\mathrm{pos}}(\boldsymbol{x}, y) = -\log \frac{\mathcal{T}_k \{\exp(\mathrm{sim}(\boldsymbol{z}_h^{\mathrm{l}}, \boldsymbol{t}_y)/T)\}}{\sum_{i=1}^{\mathrm{C}} \mathcal{T}_k \{\exp(\mathrm{sim}(\boldsymbol{z}_h^{\mathrm{l}}, \boldsymbol{t}_i)/T)\} + \sum_{i=1}^{\mathrm{N_{neg}}} \mathcal{T}_k \{\exp(\mathrm{sim}(\boldsymbol{z}_h^{\mathrm{l}}, \hat{\boldsymbol{t}}_i)/T)\}}, \quad (4)$$

where $(\boldsymbol{x}, y)$ stands for training image-label pairs, $\mathcal{T}_k(\boldsymbol{x})$ is the sum of $k$ largest elements in $\boldsymbol{x}$, $k$ is regional number and $T$ is the temperature. We use cosine similarity as the similarity metric:

$$\mathrm{sim}(\boldsymbol{t}_i, \boldsymbol{t}_j) = \frac{\boldsymbol{t}_i \cdot \boldsymbol{t}_j}{\|\boldsymbol{t}_i\| \|\boldsymbol{t}_j\|}. \quad (5)$$

Similarly, given a hard negative sample of the image denoted as $\tilde{\boldsymbol{x}}$, local negative loss is defined to make the model aware of outliers:

$$\mathcal{L}_{\mathrm{neg}}(\tilde{\boldsymbol{x}}) = -\log \frac{\sum_{i=1}^{\mathrm{N_{neg}}} \mathcal{T}_k \{\exp(\mathrm{sim}(\tilde{\boldsymbol{z}}_h^{\mathrm{l}}, \hat{\boldsymbol{t}}_i)/T)\}}{\sum_{i=1}^{\mathrm{C}} \mathcal{T}_k \{\exp(\mathrm{sim}(\tilde{\boldsymbol{z}}_h^{\mathrm{l}}, \boldsymbol{t}_i)/T)\} + \sum_{i=1}^{\mathrm{N_{neg}}} \mathcal{T}_k \{\exp(\mathrm{sim}(\tilde{\boldsymbol{z}}_h^{\mathrm{l}}, \hat{\boldsymbol{t}}_i)/T)\}}. \quad (6)$$

Intuitively, contrastive regularization forces the model to focus on ID-related regions through random crop, and keep away from outlier-related regions through negative augmentation. We demonstrate the necessity of local negative loss to make margins for unknown categories in OOD detection in Sec.5.2.

**Diversity regularization.** As local negative prompts are randomly initialized, more regularization has to be imposed on them to ensure their diversity. Therefore, in addition to the proposed contrastive regularization, we apply a diversity regularization on local negative prompts, as follows:

$$\mathcal{L}_{\mathrm{reg}} = \frac{1}{N_{\mathrm{neg}}(N_{\mathrm{neg}} - 1)/2} \sum_{1 \le i < j \le N_{\mathrm{neg}}} \mathrm{sim}(\hat{\boldsymbol{t}}_i, \hat{\boldsymbol{t}}_j), \quad (7)$$

where $N_{\mathrm{neg}}$ is the number of local negative prompts. Final Loss is a weighted sum of the above loss:

$$\mathcal{L}_{\mathrm{total}} = \mathcal{L}_{\mathrm{pos}} + \lambda_{\mathrm{neg}} \mathcal{L}_{\mathrm{neg}} + \lambda_{\mathrm{reg}} \mathcal{L}_{\mathrm{reg}}, \quad (8)$$

where $\lambda_{\mathrm{neg}}$ and $\lambda_{\mathrm{reg}}$ are the coefficients of the corresponding losses.

### 4.3 REGIONAL OOD SCORE

On consideration that regional information is especially enhanced during training, we propose Regional-MCM score to enhance OOD score beyond simple MCM and GL-MCM:

$$S_{\mathrm{R-MCM}}(\boldsymbol{x}) = S_{\mathrm{MCM}}(\boldsymbol{x}) + \mathcal{T}_k^{\mathrm{mean}} \{ \frac{\exp(\mathrm{sim}(\boldsymbol{z}_h^{\mathrm{l}}, \boldsymbol{t}_i)/T)}{\sum_{j=1}^{C} \exp(\mathrm{sim}(\boldsymbol{z}_h^{\mathrm{l}}, \boldsymbol{t}_j)/T) + \sum_{j=1}^{N_{\mathrm{neg}}} \exp(\mathrm{sim}(\boldsymbol{z}_h^{\mathrm{l}}, \hat{\boldsymbol{t}}_j)/T)} \}, \quad (9)$$

where $\mathcal{T}_k^{\mathrm{mean}}(\boldsymbol{x})$ is the mean of $k$ largest elements in $\boldsymbol{x}$ and $k$ represents regional number. Intuitively, Regional-MCM serves as a general form of GL-MCM that takes the $k$ most similar regions into consideration and additionally contains local negative prompts, which is helpful to consider more than one certain region with high similarity and improve OOD detection ability. Effectiveness of the score is verified in Sec. 5.2.

As the proposed OOD score is only applicable for OOD detection, we further propose a local-aware score that assigns a score to each class and thus determines ID category with the largest score. Specifically, ID classification takes both global and local features into consideration:

$$h_i^{\mathrm{g}}(\boldsymbol{x}) = \mathrm{sim}(\boldsymbol{z}^{\mathrm{g}}, \boldsymbol{t}_i), \quad (10)$$

Table 1: Results when ImageNet-1k is used as ID data. We compare the methods in different tuning manners. † represents the result by our re-implementation. **Bold** values represent the best results.

| Method | iNaturalist | | SUN | | Places | | Texture | | **Average** | |
|---|---|---|---|---|---|---|---|---|---|---|
| | FPR95↓ | AUROC↑ | FPR95↓ | AUROC↑ | FPR95↓ | AUROC↑ | FPR95↓ | AUROC↑ | FPR95↓ | AUROC↑ |
| *Zero-shot* | | | | | | | | | | |
| MCM | 30.91 | 94.61 | 37.59 | 92.57 | 44.69 | 89.77 | 57.77 | 86.11 | 42.74 | 90.77 |
| GL-MCM | 15.18 | 96.71 | 30.42 | 93.09 | 38.85 | 89.90 | 57.93 | 83.63 | 35.47 | 90.83 |
| *Full-tuning* | | | | | | | | | | |
| MSP | 54.05 | 87.43 | 73.37 | 78.03 | 72.98 | 78.03 | 68.85 | 79.06 | 67.31 | 80.64 |
| Energy | 29.75 | 94.68 | 53.18 | 87.33 | 56.40 | 85.60 | 51.35 | 88.00 | 47.67 | 88.90 |
| KNN | 29.17 | 94.52 | 35.62 | 92.67 | 39.61 | 91.02 | 64.35 | 85.67 | 42.19 | 90.97 |
| NPOS | 16.58 | 96.19 | 43.77 | 90.44 | 45.27 | 89.44 | 46.12 | 88.80 | 37.93 | 91.22 |
| Textual-OE | 29.61 | 94.74 | 57.12 | 87.34 | 66.82 | 83.71 | 79.29 | 77.76 | 58.21 | 85.88 |
| *Few-shot* | | | | | 4-*shot* | | | | | |
| CoOp | 18.95 | 95.52 | 29.58 | 92.90 | 38.72 | 89.64 | 48.03 | 85.87 | 33.82 | 90.98 |
| **Local-Prompt** | **9.65** | **97.87** | 20.40 | 95.57 | 29.39 | 92.67 | 51.20 | 88.00 | 27.66 | 93.53 |
| LoCoOp† | 21.67 | 95.69 | 22.98 | 95.07 | 31.41 | 92.10 | 49.79 | 87.85 | 31.46 | 92.68 |
| **Local-Prompt+LoCoOp** | 12.81 | 97.29 | **19.34** | **95.85** | **27.53** | **92.97** | **45.51** | **89.99** | **26.29** | **94.03** |
| | | | | | 16-*shot* | | | | | |
| CoOp | 14.60 | 96.62 | 28.48 | 92.65 | 36.49 | 89.98 | 43.13 | 88.03 | 30.67 | 91.82 |
| **Local-Prompt** | 8.71 | 98.10 | 23.97 | 94.85 | 32.50 | 92.32 | 47.93 | 89.04 | 28.27 | 93.58 |
| LoCoOp | 16.05 | 96.86 | 23.44 | 95.07 | 32.87 | 91.98 | 42.28 | 90.19 | 28.66 | 93.52 |
| **Local-Prompt+LoCoOp** | **8.63** | **98.07** | **23.23** | **95.12** | **31.74** | **92.42** | **34.50** | **92.29** | **24.52** | **94.48** |

$$h_i^l(\boldsymbol{x}) = \mathcal{T}_k^{\text{mean}}\{\exp(sim(\boldsymbol{z}_h^l, \boldsymbol{t}_i)/T)\}, \tag{11}$$

where $h_i^g(\boldsymbol{x})$, $h_i^l(\boldsymbol{x})$ are global and local portions and $i$ represents the category label. The score attached to each category is measured by:

$$f(y = i|\boldsymbol{x}) = h_i^g(\boldsymbol{x}) * h_i^l(\boldsymbol{x}), \tag{12}$$

each sample is then classified corresponding to the maximum score.

## 5 EXPERIMENTS

**Datasets.** Following existing works (Ming et al., 2022; Miyai et al., 2023b;a), large scale ImageNet-1K (Deng et al., 2009) along with a 100-category subset of it denoted as ImageNet-100 are used as ID dataset and OOD dataset is a combination of iNaturalist (Van Horn et al., 2018), SUN (Xiao et al., 2010), Places (Zhou et al., 2017) and Texture (Cimpoi et al., 2014). In addition, We also use two semantically similar subsets of ImageNet-1k, *i.e.*, ImageNet-10 and ImageNet-20, to evaluate near OOD detection performance (Ming et al., 2022). Detailed information is provided in Appendix A.

**Implementation details.** We use CLIP-Base/16 as the backbone. Image encoder, text encoder, and global prompts are frozen and only local prompts are learnable. We set one local prompt for each known category (1000 in total for ImageNet-1k). $\lambda_{\text{neg}}$ and $\lambda_{\text{reg}}$ are 5 and 0.5, respectively. As the stability of temperature $T$ has been verified by previous research (Ming et al., 2022; Miyai et al., 2023b), we set $T$ to be 1 by default. More details are shown in Appendix B.

**Evaluation metrics.** We report the following metrics for evaluation: (1) the area under the receiver operating characteristic curve (AUROC); (2) false positive rate of OOD samples when true positive rate of ID samples is 95% (FPR95); (3) in-distribution data classification accuracy (ID accuracy).

### 5.1 MAIN RESULTS

**ImageNet-1k as ID dataset.** We report 4 and 16-shot results on four OOD datasets using ImageNet-1k as ID dataset. It can be seen in Tab. 1 that tuning local prompts only using hand-crafted global prompts can achieve competitive results, especially in datasets with challenging fine local textures like iNaturalist. Concretely, we get impressive progress in 4-shot tuning (outperforming by 3.80% on FPR95). In 16-shot setting, our approach gets competitive results as well. All results above strongly showcase the potential of regional enhancement for OOD detection as an orthogonal direction to global prompt optimization methods. Results of more shots are shown in Appendix C.

**ID accuracy.** In addition to OOD performance, it is also meaningful to evaluate ID accuracy as it is expected to ensure classification accuracy and separate outliers at the same time. However, weighing too much about separating outliers may narrow the inter-class distance, thus harming accuracy. Consequently, we evaluate ID accuracy for ImageNet-1k and the results shown in Tab. 2 conclude that

Table 2: ID accuracy evaluation using ImageNet-1k as ID data.

| Method | Tuning Manner | ID accuracy |
|---|---|---|
| MCM | *zero-shot* | 67.01 |
| GL-MCM | | 67.01 |
| KNN | *full-tuning* | 79.64 |
| NPOS | | 79.42 |
| LoCoOp | *4-shot* | 69.50 |
| **Local-Prompt** | | **70.52** |
| LoCoOp | *16-shot* | 71.86 |
| **Local-Prompt** | | **74.24** |

Table 3: Comparison results of near OOD detection tasks. We report 16-shot results for few-shot methods. Our method can achieve performance against other methods.

| Method | ID OOD | ImageNet-10 ImageNet-20 | | ImageNet-20 ImageNet-10 | | Average | |
|---|---|---|---|---|---|---|---|
| | | FPR95↓ | AUROC↑ | FPR95↓ | AUROC↑ | FPR95↓ | AUROC↑ |
| MCM | | 5.00 | 98.71 | 12.51 | 97.70 | 8.75 | 98.21 |
| GL-MCM | | 10.10 | 98.04 | 9.00 | 98.62 | 9.55 | 98.33 |
| LoCoOp | | 11.20 | 97.49 | 12.00 | 97.79 | 11.60 | 97.64 |
| **Local-Prompt** | | **3.90** | **99.06** | **6.20** | **98.84** | **5.05** | **98.95** |

Table 4: Experiments on ImageNet-100. 4-shot results are reported for few-shot tuning.

| Method | iNaturalist | | SUN | | Places | | Texture | | Average | |
|---|---|---|---|---|---|---|---|---|---|---|
| | FPR95↓ | AUROC↑ | FPR95↓ | AUROC↑ | FPR95↓ | AUROC↑ | FPR95↓ | AUROC↑ | FPR95↓ | AUROC↑ |
| MCM | 18.13 | 96.77 | 36.45 | 94.54 | 34.52 | 94.36 | 41.22 | 92.25 | 32.58 | 94.48 |
| LoCoOp[†] | 44.94 | 93.31 | 30.70 | 94.70 | 34.31 | 93.93 | 55.92 | 90.01 | 41.47 | 92.99 |
| **Local-Prompt** | **6.99** | **98.05** | **24.34** | **96.03** | **26.04** | **95.49** | **36.32** | **93.67** | **23.42** | **95.81** |

our method significantly surpasses previous methods (1.02% and 2.38%, respectively). It demonstrates that the proposed regional enhancement is also beneficial to improve OOD separation while increasing inter-class distance, which is of great significance for real application.

**Extensive to global prompts.** Benefiting from the convenient integration with well-trained global prompts, we *combine the advantage of trained global prompts* from LoCoOp (Miyai et al., 2023a) without further design (**Local-Prompt+LoCoOp**) and get further substantial improvements on all datasets. For example, we surpass the state-of-the-art method by a large margin (5.17% on FPR95 and 1.35% on AUROC), even against previous methods with 16 shots. Specifically, extending local prompts to global prompts is helpful to datasets with major global representations like Texture, verifying the compatibility and potential of local prompt optimization. It is notable that the extension process is seamless and can be integrated with any global optimization prompts, strongly showing the extensibility of our approach. Density map in Fig. 3 also illustrates the superiority of our method. More visualization can be found in Appendix D.

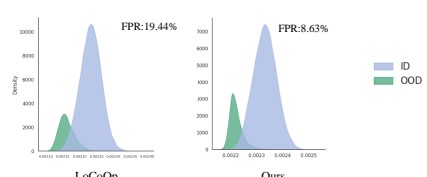

Figure 3: Comparison of density map on iNaturalist. Ours are more separable.

**Near OOD detection tasks.** Near OOD task typically evaluates semantically similar categories and is relatively more challenging. Following existing work (Ming et al., 2022), we conduct experiments on ImageNet-10 vs. ImageNet-20 to verify the effectiveness of the proposed approach. Results in Tab. 3 indicate that our method exceeds previous methods by a large margin. It can be observed that previous few-shot method fails in near OOD tasks. We attribute it to the refinement of local prompts by local outlier knowledge, which greatly assists in distinguishing semantically similar categories. By contrast, relying solely on global prompts may confuse the distribution. Therefore, it underlines that our local outlier enhancement approach is especially effective in semantically near OOD tasks.

**ImageNet-100 as ID dataset.** We also conduct experiments using ImageNet-100 as ID dataset. We compare with other zero-shot and few-shot methods and results are reported in Tab. 4. The substantial performance improvements (average 9.16% reduction on FPR95 and 1.33% promotion on AUROC) strongly demonstrate the effectiveness and transferability of our approach.

**Comparison with outlier methods.** NegLabel (Jiang et al., 2024) designs a novel scheme for OOD score with negative labels. It leverages real outlier information with negative labels from extensive corpus databases. This kind of knowledge helps to a great extent pointed out by OE (Hendrycks et al., 2018) and is inconsistent with real-world application, where negative categories are infinite. As can be seen from Tab. 5, we observe that **(1)** although without outlier, our model achieves better

Table 5: Comparison with NegLabel, which leverages powerful outlier samples.

| Method | iNaturalist | | SUN | | Places | | Texture | | Average | |
|---|---|---|---|---|---|---|---|---|---|---|
| | FPR95↓ | AUROC↑ | FPR95↓ | AUROC↑ | FPR95↓ | AUROC↑ | FPR95↓ | AUROC↑ | FPR95↓ | AUROC↑ |
| NegLabel (w outlier) | 1.91 | 99.49 | 20.53 | 95.49 | 35.59 | 91.64 | 43.56 | 90.22 | 25.40 | 94.21 |
| **Local-Prompt (w/o outlier)** | 8.63 | 98.07 | 23.23 | 95.12 | 31.74 | 92.42 | 34.50 | 92.29 | **24.52** | **94.48** |

Table 6: Effectiveness of each component in loss function.

| Loss | iNaturalist | | SUN | | Places | | Texture | | **Average** | |
|---|---|---|---|---|---|---|---|---|---|---|
| | FPR95↓ | AUROC↑ | FPR95↓ | AUROC↑ | FPR95↓ | AUROC↑ | FPR95↓ | AUROC↑ | FPR95↓ | AUROC↑ |
| Baseline ($\mathcal{L}_{pos}$) | 11.64 | 97.59 | 26.81 | 94.59 | 34.50 | 91.59 | 50.46 | 87.84 | 30.85 | 92.90 |
| $+\mathcal{L}_{neg}$ | 9.84 | 97.88 | 24.37 | 94.97 | 32.84 | 91.94 | 50.02 | 88.32 | 29.27 | 93.27 |
| $+\mathcal{L}_{neg} + \mathcal{L}_{reg}$ | 9.39 | 97.97 | 24.20 | 95.10 | 32.13 | 92.21 | 49.37 | 88.66 | **28.77** | **93.48** |

Table 7: Influence of different negative augmentation strategies.

| | iNaturalist | | SUN | | Places | | Texture | | **Average** | |
|---|---|---|---|---|---|---|---|---|---|---|
| | FPR95↓ | AUROC↑ | FPR95↓ | AUROC↑ | FPR95↓ | AUROC↑ | FPR95↓ | AUROC↑ | FPR95↓ | AUROC↑ |
| Baseline | 11.18 | 97.68 | 23.30 | 95.25 | 30.97 | 92.44 | 50.44 | 87.89 | 28.97 | 93.31 |
| +cutout | 17.47 | 96.64 | 26.17 | 94.73 | 34.38 | 91.84 | 54.84 | 86.88 | 33.21 | 92.52 |
| +gaussian noise | 11.32 | 97.59 | 25.70 | 94.39 | 34.99 | 91.71 | 51.06 | 88.15 | 30.76 | 92.96 |
| +random crop | 9.39 | 97.97 | 24.20 | 95.10 | 32.13 | 92.21 | 49.37 | 88.66 | **28.77** | **93.48** |

average performance against NegLabel; **(2)** our method achieves better balance between different kinds of OOD datasets, which strongly validates the effectiveness of incorporating local information and strengthens its application in diverse and infinite read-world scenarios.

## 5.2 ABLATION STUDY

We conduct various and comprehensive ablation studies to investigate the effectiveness of each component. We train a 4-shot model to showcase the function of components and hyper-parameters unless otherwise stated. More studies about training time and coefficient are shown in Appendix C.

**Effectiveness of loss components.** We conduct experiments to analyze the effectiveness of each loss component and results in Tab. 6 imply that local prompts optimization (baseline) achieves comparable results with global prompts optimization methods, strongly demonstrating the effectiveness of our proposed regional enhancement strategy. Furthermore, taking hard negative samples together with local negative prompts $\hat{t}$ to optimize OOD-considered representation is beneficial to the distinction of outliers. Outcome without regularization loss is better in FPR95 but slightly inferior in AUROC. Considering overall performance and the fact that the optimization of $\hat{t}$ relies heavily on the initialization without regularization, we reserve regularization loss for stability and convergence.

**Influence of negative augmentation strategy.** We conduct experiments to analyse the influence of negative augmentation strategy. In Tab. 7, baseline does not employ augmentation. Cutout, gaussian noise and random crop are used as augmentation, respectively. It is concluded that improper augmentation is unfavorable to detection ability. By contrast, random crop is helpful to enhance regional-related representation and distinguish outliers, which confirms our analysis above.

**Number of local negative prompts.** The number of local negative prompts $N_{neg}$ stands for local outlier knowledge the model learns from negative augmentation. Results in Tab. 8 indicate that the performance is positively related to $N_{neg}$. However, larger $N_{neg}$ requires more computational resources. Considering both computational cost and performance, the number 300 is selected.

**OOD score strategy.** We compare our proposed OOD score with existing scores (Ming et al., 2022; Miyai et al., 2023b) on 16-shots. We also remove local negative prompts in our OOD score to analyse their effectiveness. Results in Fig. 4 imply that our regional OOD score attains better performance than existing strategy, displaying the strength of regional enhancement. Moreover, the performance is further promoted when negative prompts are introduced. We attribute it to possible outlier leverage through negative prompts.

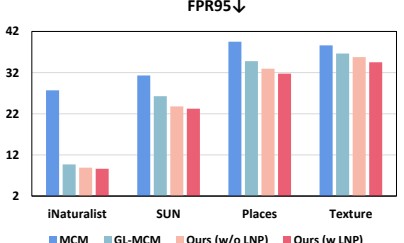 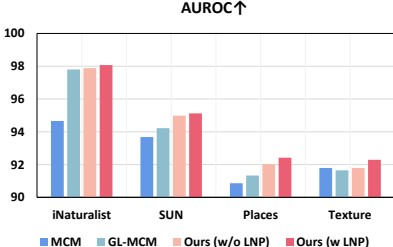

Figure 4: Ablation study of different OOD score strategies. LNP denotes local negative prompts.

Table 8: Influence of different number of local negative prompts.

| $N_{neg}$ | iNaturalist | | SUN | | Places | | Texture | | Average | |
|---|---|---|---|---|---|---|---|---|---|---|
| | FPR95↓ | AUROC↑ | FPR95↓ | AUROC↑ | FPR95↓ | AUROC↑ | FPR95↓ | AUROC↑ | FPR95↓ | AUROC↑ |
| 100 | 9.02 | 98.00 | 24.71 | 94.82 | 32.77 | 91.87 | 48.81 | 88.50 | 28.82 | 93.30 |
| 200 | 9.65 | 97.90 | 22.58 | 95.26 | 32.03 | 92.18 | 49.85 | 88.00 | 28.52 | 93.33 |
| 300 | 9.39 | 97.97 | 24.20 | 95.10 | 32.13 | 92.21 | 49.37 | 88.66 | 28.77 | 93.48 |

## 5.3 VISUALIZATION

**Visualization of local prompts.** We give a detailed explanation of local prompts and visualize the regions that attract attention from local prompts and local negative prompts. Visualization is shown in Fig. 5 in the order of original picture, ID-related, and several typical OOD-related areas. It implies that local prompts concentrate on ID regions and local negative prompts are helpful to extract ID-irrelevant, *i.e.*, outlier areas corresponding to certain semantics. For instance, local negative prompts successfully focus on sky, sea, and beach, respectively in pirate ship, which is beneficial to enhance OOD detection ability through local and unknown knowledge leverage.

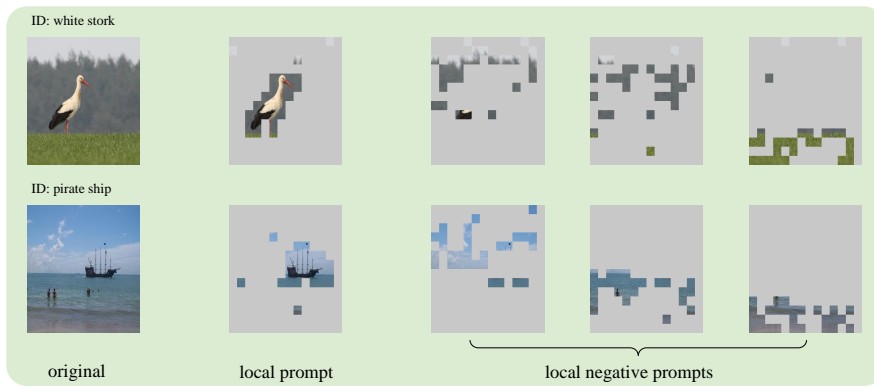

Figure 5: Visualization of ID and OOD related regions. Local prompts and local negative prompts successfully focus on ID-related and OOD-related regions, respectively, which helps OOD detection.

**Visualization of hard OOD samples.** We showcase several hard OOD samples that previous global-based method fails to detect, while our method with local enhancement successfully discriminates the outlier. For example, in the first image, previous global-based method fails to detect the outlier as it is similar to ID category ocean liner on the whole, with only subtle difference mechanical devices on the deck indicating that it is actually an icebreaker (also demonstrated by the iceberg next to it). The same phenomenon are also observed other examples.

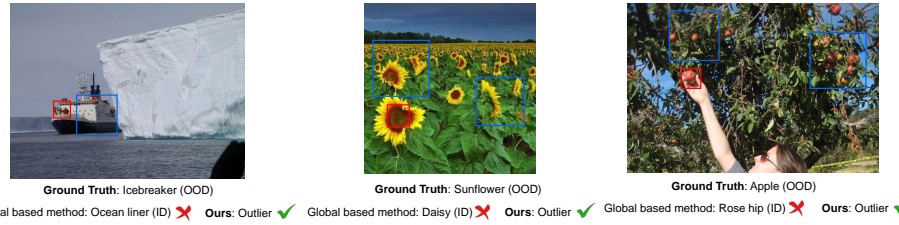

Figure 6: Examples of outlier from SUN dataset that fails to be detected by previous global-based method, and our method with local enhancement successfully discriminates them with subtle differences in certain regions.

## 6 CONCLUSION

In this paper, we investigate the relationship between global and local prompts for few-shot OOD detection. Concentrating on enhancing outlier knowledge for better local textual representation utilization, we decompose global and local features using respective prompts. We use straightforward negative augmentation to leverage local outlier knowledge and propose local prompts to enhance regional information by means of regional regularization. We also propose regional-related evaluation metric to enrich the detector with local outlier information. Moreover, our approach is extensible to well-trained global prompts to get better performance. We conduct various experiments and superior results underline the effectiveness of our method. We hope our method can provide a new perspective on studying local prompts for OOD detection tasks based on vision-language models.

ACKNOWLEDGEMENT

This work is supported by National Science and Technology Major Project (2022ZD0116500), National Natural Science Foundation of China (62222609, 62076236), and CAS Project for Young Scientists in Basic Research (YSBR-083).

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

APPENDIX

## A  DATASET DETAILS

**ImageNet-10** contains 10 classes of ImageNet-1k, including military aircraft (n04552348), sports car (n04285008), brambling (n01530575), Siamese cat (n02123597), impala (n02422699), Greater Swiss Mountain Dog (n02107574), American bullfrog (n01641577), garbage truck (n03417042), common sorrel horse (n02389026) and container ship (n03095699).

**ImageNet-20** is a 20-class subset in ImageNet-1k. Concretely, they are schooner (n04147183), canoe (n02951358), balloon (n02782093), tank (n04389033), missile (n03773504), high-speed train (n02917067), starfish (n02317335), spotted salamander (n01632458), smooth newt (n0163067 0), newt (n01631663), zebra (n02391049), European green lizard (n01693334), Nile crocodile (n01 697457), Arctic fox (n02120079), grey wolf (n02114367), brown bear (n02132136), moped (n0378 5016), steam locomotive (n04310018), space shuttle (n04266014) and snowmobile (n04252077).

**ImageNet-100** is the same 100-class subset of ImageNet-1k as MCM (Ming et al., 2022) to have a fair comparison with previous vision-language based OOD detection methods. Detail categories are provided in https://github.com/deeplearning-wisc/MCM.

## B  TRAINING DETAILS

Number of learnable context words is 16. We use SGD optimizer to train the model with 30 epochs. Learning rate is $2 \times 10^{-3}$ with a cosine schedule and batch size is 256. $m$ is set to 24 by default. We empirically set positive/negative augmentation to be 8/1, which are discussed thoroughly below. Training and testing $k$ are 50 and 10, respectively in main results. We use one single NVIDIA A6000 to run all experiments. The reported results are averaged for three runs. Extra experimental results are shown below.

## C  MORE EXPERIMENTAL RESULTS

**Experiments of different shots.** We use ImageNet-1k as ID dataset in the main experiment. For training, we use different shots, *i.e.*, numbers of images from each class, following few-shot setting. For evaluation, we use validation set of ImageNet-1k, which is composed of 50000 images from 1000 categories. Results are shown in Tab. 9.

**Influence of coefficient.** Both $\lambda_{\text{neg}}$ and $\lambda_{\text{reg}}$ determine the importance of the corresponding loss. We report average FPR95 and AUROC to show the trend when the coefficients vary. The results in Tab. 10 indicate that the performance tends to be stable when the coefficients are in a wide range. During the entire ablation process, the results are relatively robust when the ratio of the two corresponding coefficients is approximately within $[2, 10]$, with performance drop under extreme cases ($\lambda_{\text{neg}} = 0.1$ and 50). For $\lambda_{\text{neg}} = 50$, the loss for local prompts is neglected, resulting in severe

Table 9: More few-shot experimental results on ImageNet-1k.

| Method | iNaturalist | | SUN | | Places | | Texture | | **Average** | |
|---|---|---|---|---|---|---|---|---|---|---|
| | FPR95↓ | AUROC↑ | FPR95↓ | AUROC↑ | FPR95↓ | AUROC↑ | FPR95↓ | AUROC↑ | FPR95↓ | AUROC↑ |
| | | | | | 4-*shot* | | | | | |
| CoOp | 18.95 | 95.52 | 29.58 | 92.90 | 38.72 | 89.64 | 48.03 | 85.87 | 33.82 | 90.98 |
| LoCoOp | 21.67 | 95.69 | 22.98 | 95.07 | 31.41 | 92.10 | 49.79 | 87.85 | 31.46 | 92.68 |
| **Local-Prompt** | 9.65 | 97.87 | 20.40 | 95.57 | 29.39 | 92.67 | 51.20 | 88.00 | 27.66 | 93.53 |
| | | | | | 8-*shot* | | | | | |
| CoOp | 15.23 | 96.69 | 27.78 | 93.08 | 35.93 | 90.22 | 48.26 | 85.91 | 31.80 | 91.47 |
| LoCoOp | 16.34 | 96.47 | 22.40 | 94.96 | 31.86 | 91.83 | 42.20 | 89.81 | 28.20 | 93.27 |
| **Local-Prompt** | 10.17 | 97.83 | 19.82 | 95.42 | 30.83 | 92.54 | 48.26 | 88.47 | 27.27 | 93.56 |
| | | | | | 16-*shot* | | | | | |
| CoOp | 14.60 | 96.62 | 28.48 | 92.65 | 36.49 | 89.98 | 43.13 | 88.03 | 30.67 | 91.82 |
| LoCoOp | 16.05 | 96.86 | 23.44 | 95.07 | 32.87 | 91.98 | 42.28 | 90.19 | 28.66 | 93.52 |
| **Local-Prompt** | 8.71 | 98.10 | 23.97 | 94.85 | 32.50 | 92.32 | 47.93 | 89.04 | 28.27 | 93.58 |

Table 10: Influence of different coefficient in loss function.

| | | iNaturalist | | SUN | | Places | | Texture | | **Average** | |
|---|---|---|---|---|---|---|---|---|---|---|---|
| | | FPR95↓ | AUROC↑ | FPR95↓ | AUROC↑ | FPR95↓ | AUROC↑ | FPR95↓ | AUROC↑ | FPR95↓ | AUROC↑ |
| | 0.1 | 17.64 | 95.97 | 27.18 | 94.76 | 34.45 | 91.41 | 48.82 | 86.89 | 32.02 | 92.25 |
| | 2 | 9.47 | 97.97 | 24.11 | 94.99 | 32.28 | 92.07 | 47.94 | 88.86 | 28.45 | 93.47 |
| $\lambda_{\text{neg}}$ | 5 | 9.39 | 97.97 | 24.20 | 95.10 | 32.13 | 92.21 | 49.37 | 88.66 | 28.77 | 93.48 |
| | 10 | 10.19 | 97.83 | 23.46 | 95.26 | 31.59 | 92.31 | 49.57 | 88.81 | 28.70 | 93.55 |
| | 50 | 34.09 | 92.06 | 42.34 | 88.63 | 50.88 | 83.52 | 62.47 | 85.70 | 47.44 | 87.48 |
| | 0.2 | 9.83 | 97.88 | 24.60 | 94.92 | 32.65 | 92.10 | 49.82 | 88.63 | 29.22 | 93.38 |
| $\lambda_{\text{reg}}$ | 0.5 | 9.39 | 97.97 | 24.20 | 95.10 | 32.13 | 92.21 | 49.37 | 88.66 | 28.77 | 93.48 |
| | 1 | 11.24 | 97.67 | 25.87 | 94.77 | 33.54 | 91.89 | 48.38 | 88.97 | 29.75 | 93.32 |

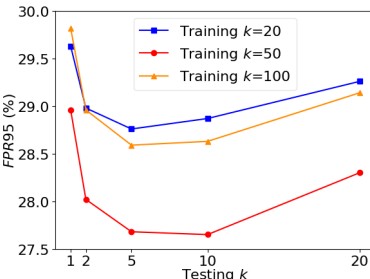
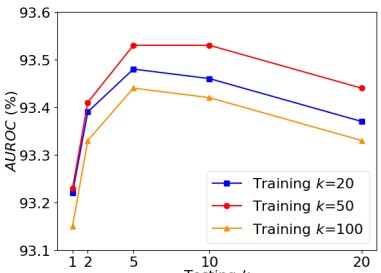

Figure 7: Results on different values of $k$ for training and inference.

performance drop. For $\lambda_{\text{neg}} = 0.1$, the local negative prompts are over-regularized by diversity constraints. The effectiveness of the proposed loss can be further verified accordingly.

**Regional number.** Regional number $k$ expresses the extent to which local prompts pay attention to areas. As regional information is not always ID or OOD-related, the selection of regional number $k$ largely determines performance. During training, Regions are selected to emphasize suitable ID-related regions (outlier regions when hard negative images) and learn regional-related local prompts. During inference, $k$ is relatively smaller to guarantee the confidence of the selected region for metric calculation. We conduct various experiments on different values of $k$ during both training and testing. Curves shown in Fig. 7 illustrate that for both $k$ during training and inference, larger $k$ will confuse confidence of the boundary between ID and OOD regions, while smaller $k$ fails to extract precise regional-related textual representations, which corresponds to our analysis. We underline that varying $k$ brings in subtle differences and even simplified outcome without careful selection of $k$ still achieves better results than previous approaches.

**Extra training time.** We extend the training time to analyse the benefits of longer training. It can be seen in Tab. 11 that 30 epochs is sufficient to get superior representations in few-shot prompt tuning and longer training time does not necessarily bring about the improvements of performance. By contrast, LoCoOp (Miyai et al., 2023a) requires 50 epochs and it indicates that our approach is well and fast converged, which is in line with the purpose of few-shot learning.

**Number of negative augmentation.** The number of negative augmentation influences the performance. It can be seen in Tab. 12 that setting negative augmentation to be 1 achieves obviously better results. A potential reason for the phenomenon is that random crops of original images are hard enough for the detector to learn local outlier information and too much negative augmentation may confuse the learning process. Consequently, we choose the least similar image as the negative augmentation.

Table 11: Influence of longer training time.

| **epoch** | iNaturalist | | SUN | | Places | | Texture | | **Average** | |
|---|---|---|---|---|---|---|---|---|---|---|
| | FPR95↓ | AUROC↑ | FPR95↓ | AUROC↑ | FPR95↓ | AUROC↑ | FPR95↓ | AUROC↑ | FPR95↓ | AUROC↑ |
| 30 | 9.39 | 97.97 | 24.20 | 95.10 | 32.13 | 92.21 | 49.37 | 88.66 | 28.77 | 93.48 |
| 50 | 9.60 | 97.83 | 23.71 | 95.05 | 32.31 | 92.02 | 48.47 | 88.68 | 28.52 | 93.40 |

Table 12: Influence of negative augmentation number.

| N | iNaturalist | | SUN | | Places | | Texture | | **Average** | |
|---|---|---|---|---|---|---|---|---|---|---|
| | FPR95↓ | AUROC↑ | FPR95↓ | AUROC↑ | FPR95↓ | AUROC↑ | FPR95↓ | AUROC↑ | FPR95↓ | AUROC↑ |
| 1 | 9.39 | 97.97 | 24.20 | 95.10 | 32.13 | 92.21 | 49.37 | 88.66 | 28.77 | 93.48 |
| 8 | 10.31 | 97.80 | 25.18 | 94.96 | 33.36 | 92.06 | 51.32 | 87.64 | 30.04 | 93.12 |

Table 13: Comparison with ID-like.

| Method | iNaturalist | | SUN | | Places | | Texture | | **Average** | |
|---|---|---|---|---|---|---|---|---|---|---|
| | FPR95↓ | AUROC↑ | FPR95↓ | AUROC↑ | FPR95↓ | AUROC↑ | FPR95↓ | AUROC↑ | FPR95↓ | AUROC↑ |
| ID-like | 8.98 | 98.19 | 42.03 | 91.64 | 44.00 | 90.57 | 25.27 | 94.32 | 30.07 | 93.68 |
| **Ours** | 12.81 | 97.29 | 19.34 | 95.85 | 27.53 | 92.97 | 45.51 | 89.99 | **26.29** | **94.03** |

**Comparison with ID-like.** We additionally compare with ID-like (Bai et al., 2024), and the results shown in Tab. 13 reveal that our model achieves better average performance against ID-like, demonstrating the utility of our local information enhancement strategy.

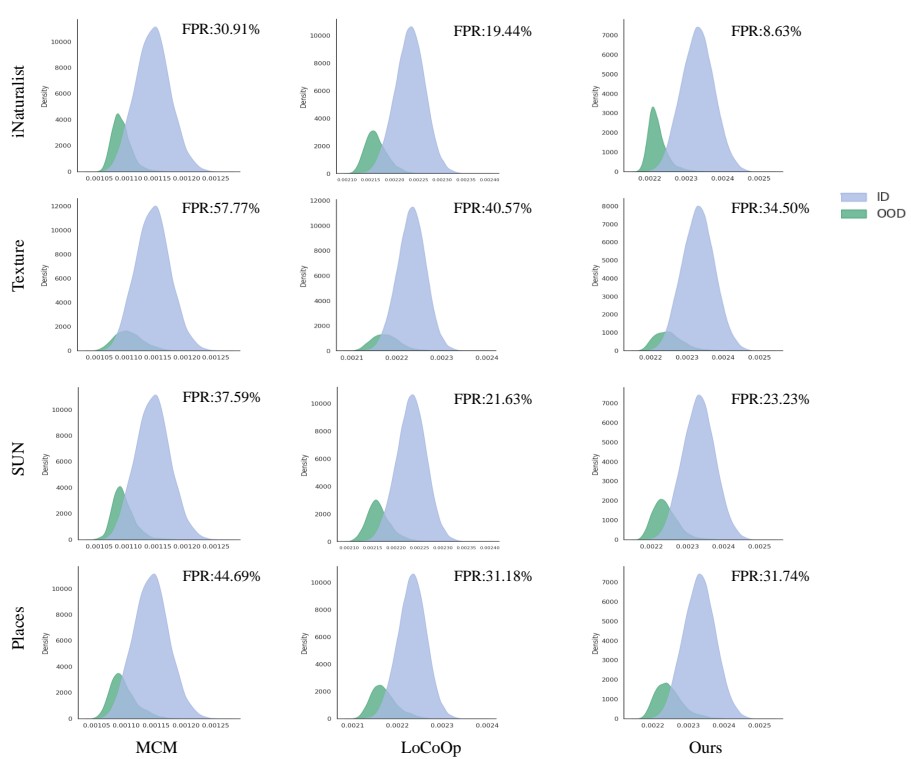

Figure 8: Density map of different OOD datasets measured by various methods.

# D VISUALIZATION AND ANALYSIS

**Visualization of heatmaps.** we additionally include heatmaps maps of typical images to illustrate the attention of our methods. It can be seen in Fig. 9 that our method accurately focuses on ID regions despite the interference of complex background surroundings, *e.g.*, tabby cat, lion and leonberg with complex background of grassland, which showcases the effectiveness of pseudo background augmentation serving as local outlier. Moreover, our method also captures multi-objects, *e.g.*, the ships on the sea (third picture on the first row), strongly validating the usefulness of local enhancement in the framework.

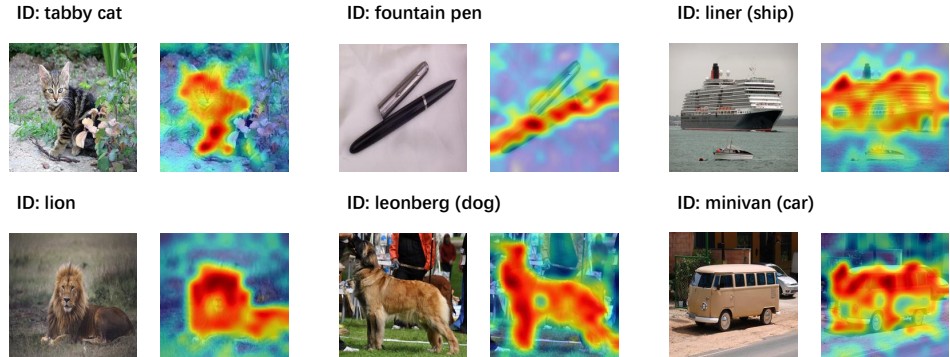

Figure 9: Visualization of heatmaps.

Table 14: The top-k average similarity of both ID-dataset and OOD datasets with local prompts.

| $k$ | 10 | 20 | 50 |
|---|---|---|---|
| w/o local enhancement | | | |
| ID-dataset | 0.297 | 0.286 | 0.272 |
| OOD-datasets | 0.280 | 0.276 | 0.261 |
| w local enhancement | | | |
| ID-dataset | 0.312 | 0.306 | 0.282 |
| OOD-datasets | 0.273 | 0.266 | 0.249 |

**More visualization.** We visualize the density map of 4 OOD datasets, *i.e.*, iNaturalist, SUN, Places and Texture and compare them with typical existing methods. Fig. 8 exhibits that our approach is capable of distinguishing ID and OOD samples with more confidence and has obviously less overlap between them than previous zero-shot and few-shot methods. It strongly demonstrates the effectiveness and potential of our local prompt based method.

**Similarity of enhanced regions.** We calculate the top-k average similarity of both ID-dataset and OOD datasets with local prompts to qualitatively showcase the effectiveness of local enhancement. It can be clearly seen in Tab. 14 that the method without local enhancement has no obvious boundary between ID and OOD datasets (0.272 for top-50 and 0.276 for top-20), which may confuse the model when detecting hard outlier samples where subtle difference occurs in certain regions of the image. By contrast, our method appears to be more discriminant and has an obvious boundary between them (0.282 for ID and 0.273 for OOD). It additionally demonstrates the core problem that current method encounters and our method of refining local regions successfully addresses the issue to some extent.

# E    LIMITATIONS AND FUTURE WORK

**Training time.** The primary limitation of our work is that it may require more training time due to negative augmentation. However, as few shot tuning paradigm for vision-language model is efficient and time-saving, it will not be the bottleneck of the model. Practically, we fine-tune CLIP on one single NVIDIA A6000 and 4-shot tuning only needs 4 hours.

**Joint training of both global and local prompts.** In our work, we mainly concentrate on the effectiveness of local prompts as an enhancement and set global prompts frozen/from existing methods. We believe that exploiting the connection between global and local features is promising. Therefore, we are in an effort to explore the advantage of training global and local prompts jointly.

**Expansion to other recognition tasks.** In this work, we focus on OOD detection for classification task. It will be an interesting topic to handle other vision tasks in open-world such as segmentation and object detection. However, existing methods also merely deal with one task at a time. Nevertheless, we believe the proposed local prompts can be generally useful as long as the applied tasks have rich local information.

