# OpenReview forum: "Local-Prompt: Extensible Local Prompts for Few-Shot Out-of-Distribution Detection"
_ICLR.cc/2025/Conference — ICLR 2025 Poster_

### Official Review · Reviewer_GNZR · 2024-11-01

**Soundness:** 3
**Presentation:** 3
**Contribution:** 3
**Rating:** 6
**Confidence:** 4

**Summary:**

The authors propose an approach to Out-of-Distribution (OOD) detection in Vision Language Models through few-shot fine-tuning. The paper uses random crops from the images to generate negative global prompts, which are used to guide the learning of learnable local prompts aimed at detecting local or regional outliers. They also propose a metric for quantifying the OOD scores that takes into account local or regional information.

**Strengths:**

I think the paper sheds light on an interesting perspective regarding the role of local or regional features in OOD detection. The idea of randomly cropping the image and creating negative global samples based on image-text similarity scores, while simple, is both effective and clever. The authors have empirically demonstrated the effectiveness of learning local prompts, achieving strong performance.

**Weaknesses:**

I am unsure about the justification for using few-shot fine-tuning. I do not understand what the authors meant in lines 46-48. Having access to the full dataset for a class can provide a better estimate of the outliers. Identifying outliers with only a few examples, while challenging, may be biased toward the examples of the class being used, especially if the number of shots is low and one of the examples is an outlier. The reason I am focusing on this is that the results in Table 1 seem counterintuitive to me. It appears that OOD detection worsens with 16 shots on the SUN and Places datasets, where we would expect the model to identify OODs better with more examples.

There also seems to be a lack of ablation studies regarding the model's performance without the learnable negative local prompts. I believe this ablation is important.

I have a comment about Figure 2. It mentions local features of augmented inputs, which I believe refers to the image patches from the randomly cropped images. However, from the figure, it appears that they are only taking patches from the entire image, which is confusing.

**Questions:**

I kindly request that the authors clarify the motivation behind few-shot tuning and address the discrepancy between few-shot and many-shot performance in Table 1, as highlighted in the weaknesses section. I believe an ablation study on the impact of additional negative local prompts is necessary to justify their use. Additionally, I would appreciate it if the authors could provide a clearer explanation of the image in Figure 2.

---

> ### Author Response · Authors · 2024-11-22
> **Author Response to Reviewer GNZR for Q1-Q2**
>
> We appreciate the reviewer for the valuable comments and appropriately respond to them as follows. We will add the suggested experiments and illustrations in the revised version.
>
> #### **Q1:  Justification for using few-shot fine-tuning.**
> A1: Few-shot OOD detection is proposed by LoCoOp[3]. We agree with their settings and introduce them briefly in the introduction. Here we provide the explanation in detail.
>
> "On the one hand, the zero-shot methods do not require any training data, but they may encounter a domain gap with ID downstream data, which limits the performance of zero-shot methods[4]. On the other hand, fully supervised methods utilize the entire ID training data but may destroy the rich representations of CLIP by fine-tuning, which also limits the performance despite requiring enormous training costs. To overcome the limitations of fully supervised and zero-shot methods, it is essential to develop a few-shot OOD detection method that utilizes a few ID training images for OOD detection. By finding a balance between the zero-shot and fine-tuned methods, few-shot OOD detection has the potential to provide a more efficient and effective solution for OOD detection."
>
> The effectiveness and efficiency have been demonstrated by previous works[3,5], and few-shot OOD detection has the potential to get competitive results against full-tuning methods[6,7]
>
> For the reviewer's concern that the performance may be biased toward the examples of the class being used, in the paper, we conduct standard processing of few-shot selection following existing methods in both representation learning[1,2] and OOD detection[3], and select few-shot samples using exactly the same procedure. Therefore, all comparisons are conducted under the same setting. As for the performance, we are not sure about the random seed used by previous works. In our work, all of the results are averaged for three runs and we do not observe obvious fluctuations.
>
> We hope the analysis could help the reviewer have a better understanding of few-shot OOD detection. If you still have concerns or other questions, respond to us and we will make in every effort to address your concerns.
>
>
> #### **Q2:  Few-shot and many-shot performance.**
> A2: For OOD detection performance on SUN and Places datasets, on the one hand, the phenomenon is common under the setting of few-shot OOD detection, with **similar trend is observed** in ID-like[5] (1-shot and 4-shot in Places, shown in table below) and LoCoOp[4] (4 shot and 16 shot in SUN and Places shown in Table.1). Possible reasons for the phonomenon are: **1)** the ability of few-shot learning for effectively gaining outlier knowledge and enhancing OOD detection performance with scarce samples, certificating the utility; **2)** Moreover, for few-shot OOD detection, more samples may not necessarily bring better performance, and incorporating more samples may confuse the discrimination, which should **be the characteristic of the two datasets**.
>
> |method|iNaturalist||SUN||Places||Texture||
> |-|-|-|-|-|-|-|-|-|
> ||FPR|AUORC|FPR|AUORC|FPR|AUORC|FPR|AUORC|FPR|AUORC
> |ID-like (1 shot)|14.57 |97.35|44.02 |91.08| 41.74 |91.15 |26.77| 94.38|
> |ID-like (4 shot)| 8.98 |98.19|  42.03 |91.64| 44.00| 90.57 |25.27 |94.32| 26.08| 94.36|

---

> ### Author Response · Authors · 2024-11-22
> **Author Response to Reviewer GNZR for Q3-Q4**
>
> #### **Q3: Performance without learnable negative local prompts.**
> A3: As the negative local prompts represent potential outlier knowledge, they are randomly initialized since we have no access to outlier samples during training, therefore it can not be replaced with hand-crafted prompts in the method. So we assume the reviewer hopes to remove the negative local prompts along with hard-negative samples (which are used to optimize the negative local prompts). The results are shown in Fig.4 termed as w/o LNP. We showcase the detailed results as follows:
>
> |method|iNaturalist||SUN||Places||Texture||Average||
> |-|-|-|-|-|-|-|-|-|-|-|
> ||FPR|AUORC|FPR|AUORC|FPR|AUORC|FPR|AUORC|FPR|AUORC|FPR|AUORC|
> |Ours (w/o LNP)|8.91|97.88|23.78|94.98|32.94|92.02|35.76|91.79|25.34|94.16|
> |Ours (w LNP)|8.63|98.07|23.23|95.12|31.74|92.42|34.50|92.29|**24.52**|**94.48**|
>
> It can be seen that with negative local prompts, the model gets consistent and substantial improvements, showcasing the effectiveness of negative local prompts for enhancing local outlier knowledge with respect to few-shot OOD detection.
>
> As for the qualitative effectiveness of the negative local prompts, we provide additional visualization in Fig.5 to demonstrate that they actually learn outlier knowledge in different scenarios.
>
> #### **Q4: Question about Figure.2 .**
> A4: We sincerely appreciate your raising this point. First, we clarify that augmented inputs indeed **refer to the image patches from the randomly cropped images (marked with yellow in the top diagram of Fig.2)**, which is in line with your understanding. In the bottom diagram of Fig.2, we take one cropped image as an example, and the orange patches are local features from the same augmented image. For loss calculation and evaluation, the most related regions (patches after regional selection in the diagram) are selected for enhancing OOD detection performance. We are grateful for the reviewer's rigorous reading, and hope the explanation could address your concerns.
>
> We are truly grateful for the reviewer's interest in our methods with detailed reviewing, and hope the explanation could address your concerns. If you have any further questions, reply to us at your convenience and we will try our best to address your concern.
>
>
>
> [1] Learning to Prompt for Vision-Language Models. IJCV 2022.
>
> [2] Conditional Prompt Learning for Vision-Language Models. CVPR 2022.
>
> [3] LoCoOp: Few-Shot Out-of-Distribution Detection via Prompt Learning. NeurIPS 2023.
>
> [4] Delving into out-of-distribution detection with vision-language representations. NeurIPS 2022.
>
> [5] ID-like Prompt Learning for Few-Shot Out-of-Distribution Detection. CVPR 2024.
>
> [6] Energy-based Out-of-distribution Detection. NeurIPS 2020.
>
> [7] Non-Parametric Outlier Synthesis. ICLR 2023.

---

> ### Author Response · Authors · 2024-11-25
> **Looking forward to your feedback**
>
> Dear reviewer GNZR:
>
> We respectfully appreciate again for your insightful and thoughtful comments! As the suggestions from the reviewer, we give thorough explanation and update the manuscripts accordingly.
>
> As it is approaching the end of the discussion period (November 26 at 11:59 pm AoE) and we do not receive feedback, we sincerely hope you could look through our response and have a further comment at your convenience if you have any questions about the paper. We will do our best to address the issues of the reviewer.
>
> Best wishes,
>
> Submission 6694 Authors.

---

> ### Comment · Reviewer_GNZR · 2024-11-25
> **Final Rating**
>
> I think the authors addressed most of my concerns and I would like to stick to my previous rating of marginally above acceptance. It has some interesting observations.
>
> Couple of comments:
>
> - I don't think I understand authors' point of adding more shots causing more confusion for OOD detection, because intuitively having more shots should make the model more confident about in distribution data.
>
> - The improvement from negative local prompt seems to be marginal, despite being a key selling point of the paper. Is there any intuition why?

---

> ### Author Response · Authors · 2024-11-26
> **Additional Response to Reviewer GNZR**
>
> We sincerely thank the reviewer for taking the time to view the rebuttal and making valuable suggestions to the paper. As for the two additional questions, we would like to make some further clarifications:
>
> #### **1. Adding more shots**
> As can be seen from the main table and the explanation in the first-round discussion, we would like to emphasize two key points: **1)** the phenomenon is commonly observed in various OOD detection methods, including but not limited to ID-like and LoCoOp, this part we have adequately showcase in the first round discussion. **2)** the phenomenon should be attributed to the characteristics of the two datasets.
>
> To further support our point of view, we calculate the similarity of prompts between both OOD datasets and ID-dataset with different shots and observe the trends of change:
>
>
> |Average similarity|4-shot|16-shot|
> |-|-|-|
> |ID-dataset|0.297|0.304|
> |iNaturalist|0.264|0.255|
> |SUN|0.282|0.281|
> |Places|0.285|0.283|
> |Texture|0.289|0.283|
>
> It can be seen from the table that **1)** the similarity of the four OOD datasets are related to the OOD detection performance on them. Specifically, more similar to ID datasets (Texture) leads to poor performance, which is in line with the analysis and experimental results; **2)** as the number of shot increases, the discrimination between SUN and Places (0.01 and 0.02 decrease) is vaguer than that in iNaturalist and Texture (0.09 and 0.06 decrease, respectively). It partially proves that the gain brought by more shots from the two middle datasets is smaller. It is the characteristic of the two datasets, and is **irrelevant to the algorithm used for OOD detection**.
>
>
> From the analysis above, we would like to explain that **we are not the only one that encounters the phenomenon**, and we **give possible explanation from our perspective of view** to showcase that the phenomenon does exist other than the bias brought by few-shot setting, which indicates the rationality of the phenomenon. As it is not the focus of our paper, **the effectiveness of our proposed method that our proposed method achieves consistent and substantial improvements should not be ignored**. The reviewer raises an interesting point and we will take investigation of balance between different kinds of datasets as future research points.
>
>
> If the reviewer still has concerns about the phenomenon, point out in detail which part of the explanation above you have questions, and we are pleased to make further explanation.
>
> #### **2. Improvement from negative local prompt**
> We would like to emphasize that negative local prompt is **one core contribution of the proposed method** and our contributions can be concluded as:
> **1)** utilization of local prompts (enhance local information), **2)** local negative prompts (provide outlier knowledge) and **3)** local based OOD metrics. We give a detailed comparison of the proposed method with LoCoOp in a progressive manner to demonstrate the effectiveness of each component.
>
> |method|iNaturalist||SUN||Places||Texture||Average||$\Delta$|
> |-|-|-|-|-|-|-|-|-|-|-|-|
> ||FPR|AUORC|FPR|AUORC|FPR|AUORC|FPR|AUORC|FPR|AUORC|
> |LoCoOp_{GL}|21.67| 95.69 |22.98 |95.07 |31.41| 92.10 |49.79| 87.85 |31.46 |92.68||
> |Ours_{GL}|9.69|97.80|26.27|94.22|34.78|91.33|36.63|91.65|26.84| 93.75|**4.62%**|
> |Ours_{R-MCM}(w/o LNP)|8.91|97.88|23.78|94.98|32.94|92.02|35.76|91.79|25.34|94.16|**6.12%**|
> |Ours_{R-MCM}(w LNP)|8.63|98.07|23.23|95.12|31.74|92.42|34.50|92.29|**24.52**|**94.48**|**6.94%**|
>
>
>
> We summarize from the table that **1)** compared with LoCoOp (with the same OOD metric), we get a substantial improvement (**4.62%** FPR95), which firmly validates the effectiveness of local prompts; **2)** our method equipped with local and outlier enhancement further improves the performance by **2.32%** on FPR95. We shall say all these improvements are made by our local enhancement framework and should not be ignored.
>
>
> Last but not least, please allow us to say that improvement on few-shot OOD detection is challenging. Given that we already substantially improve the performance by about 4.62% on FPR95 compared to LoCoOp with the same OOD metric, which is approximately the promotion from 4-shot to 16-shot, such **an improvement building on several methods that proposed by us, i.e., ours_{GL} to ours_{w LNP} should not be considered to be marginal**.
>
>
> We really appreciate the reviewer's patience and valuable time to view the rebuttal, and rates a positive points for the paper. We hope the additional response could address the remaining questions from the reviewer. We are more than pleased to discuss with the reviewer about any of the problems and discover some observations that are worth paying attention to.

---

### Official Review · Reviewer_KX2t · 2024-11-02

**Soundness:** 2
**Presentation:** 3
**Contribution:** 2
**Rating:** 6
**Confidence:** 4

**Summary:**

This work proposes to enhance Out-of-Distribution (OOD) detection through local prompt learning. Specifically, the method involves randomly cropping training images and identifying hard negatives by comparing their similarity to global prompts, which are provided in textual format. These hard negatives, along with the associated object samples, are then used to train local negative prompts.

**Strengths:**

1. The paper is well-written overall, but it lacks some crucial symbol definitions and implementation details.
2. The method addresses a significant task aimed at detecting OOD images using only a few images from the ID data. Extensive experiments demonstrate the effectiveness of the method.

**Weaknesses:**

1. Some crucial symbol definitions, such as T_k and ​\hat{t}_i , are not clearly explained.
2. Apart from replacing the maximum process with an average in Equation 3, what distinguishes the proposed method from [Miyai et al. 2023b]?
3. Although hard negative samples can serve as a form of OOD data, they primarily consist of cluttered backgrounds or parts of in-distribution (ID) objects lacking distinctive features. Essentially, they are quite different from true OOD classes. The work lacks an explanation of the underlying theories.

**Questions:**

1.  How are the local prompts and negative local prompts initialized, and what are their dimensions?
2.  Moreover, from lines 270 and 237, it appears that global and local prompts t_k share the same format, "a photo of {class}." However, Figure 2 indicates that local prompts are trainable, which seems contradictory. Could the authors provide further clarification? Are the local prompts in Equation 4 simply using the same symbol t_k but representing different meanings? If so, using distinct symbols might be more appropriate.

---

> ### Author Response · Authors · 2024-11-22
> **Author Response to Reviewer KX2t for Q1-Q2**
>
> We appreciate the reviewer for the valuable suggestions and appropriately respond to them as follows. We will add the suggested experiments and explanations in the revised version.
>
> #### **Q1: Definitions of crucial symbol.**
> A1: We thank the reviewer's kind reminder, $\mathcal{T_k}(x)$ is the sum of $\textit{k}$ largest elements in $x$ (line184). In our paper, the shape of $x$ is the number of local tokens/features. $\hat{t_i}$ is the notation of local negative prompts (line266), with subscript i from 1 to number of negative local prompts $N_{neg}$ in Eqn.6 and Eqn.7. We give a detailed explanation of relevant notations in the updated manuscripts. If the reviewer still has doubts about any of the symbol definitions, we are pleased to address your questions.
>
> #### **Q2: The difference between the proposed method and GL-MCM[Miyai et al. 2023b].**
> A2: We would like to emphasize the difference between our method and GL-MCM[Miyai et al. 2023b]. First, from the motivation, GL-MCM focuses on **ID detection**, which is a totally different OOD setting (ours OOD detection as a reference). Consequently, it merely needs to detect **any of the regions that are similar to ID categories**, so the maximum process is reasonable. However, cases are totally different in the OOD detection setting, which has to detect any possible outlier regions. By contrast, the motivation of our method is to **detect hard OOD regions**. Moreover, we kindly remind that the proposed score is a small part of our method. Concretely, we propose local prompt and negative local prompts to enhance local outlier knowledge. We focus on hard OOD regions with generated hard OOD samples, and optimize the corresponding prompts with proposed loss functions to obtain outlier information.
>
> To further enhance the explanation, we conduct OOD detection using GL-MLM score, the results are shown in the table below.
>
> |method|iNaturalist||SUN||Places||Texture||Average||
> |-|-|-|-|-|-|-|-|-|-|-|
> ||FPR|AUORC|FPR|AUORC|FPR|AUORC|FPR|AUORC|FPR|AUORC|FPR|AUORC
> |GL-MCM|15.18| 96.71| 30.42| 93.09| 38.85 |89.90| 57.93| 83.63|35.47 |90.83|
> |Ours (GL-MLM)|9.69|97.80|26.27|94.22|34.78|91.33|36.63|91.65|26.84| 93.75|
> |Ours |8.63|98.07|23.23|95.12|31.74|92.42|34.50|92.29|**24.52**| **94.48**|
>
> It can be concluded that **1)** compared with GL-MCM, our method achieves a consistent and substantial promotion (10% and 3% on average, respectively), which is in line with the analysis above that GL-MCM fails to achieve satisfying results in OOD detection; **2)** our R-MCM once more gets improvement against our GL-MCM, demonstrating that the proposed method better fits for enhancing OOD detection with fine outlier knowledge.
>
> We compare the mentioned work in detail to show the unique strength and advantage of our method both theoretically and experimentally. If the reviewer still has questions, raise specific questions and we will try our best to address them.

---

> ### Author Response · Authors · 2024-11-22
> **Author Response to Reviewer KX2t for Q3-Q5**
>
> #### **Q3: Explanation of hard negative samples.**
> A3: Actually, what counts for OOD detection is that the space **is not ID regions**, so separating ID and OOD regions is of great significance in the method, and employing hard negative samples perfectly suits the requirement as carefully selected OOD data may not contribute to separating ID and OOD regions in the scene. **Theoretically**, hard negative samples have the most similar distribution with ID samples than real outliers. Once our model is trained on hard-negative samples, it naturally gains the ability to discriminate outlier samples as detecting hard negative samples (similar to ID samples) is harder than real outliers. We emphasize that the goal is not easy to realize, as we carefully design both local prompts to learn local information of ID categories, negative local prompts to simulate potential outliers, and use a diversity regularization to make negative local prompts cover more unseen classes. All the analysis above strongly demonstrates the effectiveness of the proposed method.
>
> To support our point of view,
> **1)**  **Quantitatively**, we conduct experiments about the number of negative local prompts in Table.7 and find that local negative prompt is well sufficient to achieve good results, indicating that the local negative prompts represent features of potential outliers well. We additionally carry out experiments to calculate the similarity of both ID-dataset, hard negative samples, and OOD datasets with local prompts and list qualitative results as follows:
>
> ||Average similarity|
> |-|-|
> |w/o local enhancement|
> |ID-datasets|0.297|
> |hard negative samples|0.292|
> |OOD-datasets|0.280|
> |w local enhancement (ours)|
> |ID-datasets|0.312|
> |hard negative samples|0.281|
> |OOD-datasets|0.273|
>
> It can be seen in the table that without local enhancement, the model fails to discriminate between ID-datasets and hard negative samples, and a large margin exists with OOD-datasets. After local enhancement, the difference between ID-datasets and hard negative samples obviously improves, and hard negative samples share similar similarity with OOD-datasets. The results demonstrate the effectiveness of hard negative samples and the rationality for hard negative samples serving as a form of outlier samples.
>
> **2)** **Qualitatively**, we visualize both local negative prompts in Fig.5 and attention regions of local prompts in Fig.9. It is vividly shown in these visualizations that the negative local prompt is discriminative about potential outliers in the scene (Fig.5), and pushing away from these potential outliers (e.g., background) is helpful for the model to concentrate on ID regions (Fig.9) and therefore improve the performance.
>
> We sincerely hope our analysis could help the reviewer have a better understanding of the strength of the proposed method and the rationality for hard negative samples serving as a form of outlier samples. If you still have questions, point out in detail and we will try our best to make it more clear.
>
> #### **Q4: Initialization of local prompts and negative local prompts.**
> A4: The global prompts are hand-crafted prompts "a photo of {class}" (line237) and set frozen during training and evaluation. The local prompts are initialized with embeddings of "{learnable prefix}+{class}" (line264) and optimized during training. Negative local prompts are initialized randomly due to lack of real outlier information and diversity regularization $\mathcal{L}_{reg}$ guarantees that negative local prompts cover more unseen spaces. Intuitive explanation can also be found in diagram Fig.2. Their dimensions are the same as the dimension of text embeddings, which is 512 in our paper.
>
> #### **Q5: Further clarification of prompts.**
> A5: We would like to make an explanation that $t_c$, $t$ and $\hat{t}$ stand for features of global, local and negative local prompts in our paper (defined in line 237, 264 and 265, respectively). As explained in Q4, global prompts are hand-crafted prompts in the form of "a photo of {class}" and set frozen. In practical application, local prompts are initialized with embeddings of "{learnable prefix}+{class}" and "learnable prefix" are learned in end-to-end optimization with the weighted sum of loss function. Local prompts in Eqn.4 share the same meaning with the symbol $t_k$ (k as the subscript from 1 to number of category). Intuitive explanation can also be found in diagram Fig.2. We thank the kind remind from the reviewer and make a clear explanation in the updated manuscript. If the reviewer has any further questions about the notation, point it out and we will make a clear and detailed explanation.
>
> [1] LoCoOp: Few-Shot Out-of-Distribution Detection via Prompt Learning. NeurIPS 2023.
>
> [2] Zero-Shot In-Distribution Detection in Multi-Object Settings Using Vision-Language Foundation Models.

---

> > ### Comment · Reviewer_KX2t · 2024-11-25
> >
> > Thanks for the authors’ response. The replies have addressed most of my concerns.

---

> > > ### Author Response · Authors · 2024-11-25
> > >
> > > We thank the reviewer KX2t for the time and effort in reviewing our rebuttal and we sincerely appreciate the reviewer for thorough and detailed review of our paper.
> > >
> > > We are delighted to see our paper highly recognized by the reviewer and our response addresses the concerns of the reviewer and allows the reviewer to raise the rating about the assessment of the paper!  We are more than happy to answer any concerns or questions the reviewer might still hold during the discussion period. Please do not hesitate to let us know!

---

> ### Author Response · Authors · 2024-11-25
> **Looking forward to your feedback**
>
> Dear reviewer KX2t:
>
> We respectfully appreciate again for your insightful and thoughtful comments! As the suggestions from the reviewer, we give thorough explanation and update the manuscripts accordingly.
>
> As it is approaching the end of the discussion period (November 26 at 11:59 pm AoE) and we do not receive feedback, we sincerely hope you could look through our response and have a further comment at your convenience if you have any questions about the paper. We will do our best to address the issues of the reviewer.
>
> Best wishes,
>
> Submission 6694 Authors.

---

### Official Review · Reviewer_NiJR · 2024-11-02

**Soundness:** 3
**Presentation:** 3
**Contribution:** 3
**Rating:** 6
**Confidence:** 3

**Summary:**

The paper presents a novel approach to few-shot out-of-distribution (OOD) detection. The aim of this paper is at addressing a limitation of existing few-shot OOD detection methods, which predominantly rely on global prompts and struggle to identify challenging OOD images that differ only locally from in-distribution (ID) images.

To enhance detection accuracy for these difficult cases, the proposed method introduces learnable local prompts. Specifically, the proposed method first picks out multiple randomly cropped regions from the original image and classifies them as positive or negative based on their similarity to the global text embedding. These regions are then used as tarining exampes to learn the positive and negative learnable local  prompts.

Experimental results on several popular benchmark datasets for few-shot OOD detection show that the proposed method can achieve better accuracy than several existing few-shot OOD detection methods.

**Strengths:**

- This paper proposes a novel idea of learning local prompts, which has not been well explored in the context of few-shot OOD detection.

- Experimental results reported in this paper show that the proposed method outperforms existing methods.

- This paper is well-organized and easy to follow.

**Weaknesses:**

My main concerns are about the experimental results.

**W1**. Some of the results, such as AUROC in Table 1, show very small performance differences (less than 1%) between the proposed method and the existing methods. Since the proposed method uses random cropping to learn local prompts, the standard deviation of accuracy should be reported to show the significance of the performance differences.

**W2**. Looking at Fig. 4, the accuracy differences between Ours w/ LNP and Ours w/o LNP appear to be quite small. LNP is one of the core ideas of the proposed method. If the accuracy differences are small, the superiority of the proposed method would be questionable.

**W3**. As stated in the second paragraph of the introduction section, the aim of this paper is to detect hard OOD samples that are similar to the ID classes as a whole, but can be distinguished only by looking at subtle local differences. However, there is no support in this paper for whether the proposed method can actually detect such a hard OOD sample. It would be good to show some examples of OOD images that could not be detected with the global prompts alone, but could be detected with the introduction of the local prompts.

**W4**. Based on the results in Table 9, the authors claim that the accuracy of the proposed method is not sensitive to the values of the two hyperparameters $\lambda_{\text{neg}}$ and $\lambda_{\text{reg}}$. However, this claim seems to contradict the effectiveness of the proposed method since the learning of the local prompt, which is the core of the proposed method, is controlled by these two hyperparameters (see Eq. 8).

**W5**. Table 8 shows that the FPR95 values change nonlinearly with the number of shots; the FPR95 values improve as the number of shots increases from 4-shot to 8-shot, but deteriorate as it increases to 16-shot. What is the reason for this?

**Questions:**

As noted in the Weaknesses section, I believe the experiments have several significant shortcomings. While all of these issues are important, I have listed them in descending order of priority. The first four points (**W1-W4**) directly concern the effectiveness of the proposed method and the motivation of this paper. I would like to encourage the authors to address these points in their response.

---

> ### Author Response · Authors · 2024-11-22
> **Author Response to Reviewer NiJR for Q1-Q3**
>
> We appreciate the reviewer for the valuable suggestions to our work. We reply to them sequentially and will complement in the revised version.
> #### **Q1: Standard deviation of metrics.**
> A1: All of the results are averaged for three runs and we do not observe obvious fluctuations as the variance shown in the table below. It showcases that the improvements are stable and do not attribute to the randomness. We speculate the reason is that background information of the images is rich enough to capture local outlier knowledge and thus achieve stable results, which are validated by visualization in Fig.4. Moreover, we would like to emphasize that few-shot OOD detection is challenging and the improvement is not small, which is shown in table in Q2 that our method obtains 22.04% (FPR95 )and 2.0% (AUROC) relative gain compared with LoCoOp.
>
>
> |method|iNaturalist||SUN||Places||Texture||
> |-|-|-|-|-|-|-|-|-|
> ||FPR|AUORC|FPR|AUORC|FPR|AUORC|FPR|AUORC|
> |Ours (4 shot)|12.81±0.38|97.29±0.18|19.34±0.36|95.85±0.17|27.53±0.51|92.97±0.37|45.51±0.75|89.99±0.36|
> |Ours (16 shot)|8.63±0.29|98.07±0.05|23.23±0.24|95.12±0.08|31.74±0.34|92.42±0.16|34.50±0.62|92.29±0.24|
>
>
> #### **Q2: Accuracy differences between w/ LNP and w/o LNP.**
> A2: We showcase the detailed results of w/ LNP and w/o LNP of 16 shot in the table below. We would like to highlight that the benefit margin is gradually decreasing and the improvement is actually not small. We take the improvements from 4-shot to 16-shot as a reference. Compared to w/o LNP, ours with LNP achieves **0.82** (**2.62%** relatively) and **0.32** (**0.4%** relatively) gain with respect to FPR and AUROC, respectively. By contrast, the improvements from 4 shot to 16 shot is 1.77 and 0.45, respectively as a reference. From the results, it can be seen that the improvement is not small, therefore demonstrating the effectiveness of the proposed LNP OOD metric.
>
>
> |method|iNaturalist||SUN||Places||Texture||Average||$\Delta$||
> |-|-|-|-|-|-|-|-|-|-|-|-|-|
> ||FPR|AUORC|FPR|AUORC|FPR|AUORC|FPR|AUORC|FPR|AUORC|FPR|AUORC|
> |LoCoOp|21.67|95.69 |22.98| 95.07| 31.41| 92.10 |49.79| 87.85| 31.46 |92.68|-|-|
> |Ours (w/o LNP)|8.91|97.88|23.78|94.98|32.94|92.02|35.76|91.79|25.34|94.16|19.42%|1.6%|
> |Ours (w LNP)|8.63|98.07|23.23|95.12|31.74|92.42|34.50|92.29|**24.52**|**94.48**|**22.04%**|**2.0%**|
> |Ours (4 shot )| 12.81| 97.29 |19.34 |95.85| 27.53| 92.97 |45.51| 89.99| 26.29 |94.03|
>
> #### **Q3: Examples of OOD images.**
> A3: We appreciate the reviewer for detail and thorough understanding of the proposed method. As the reviewer suggests, we showcase several examples where global based method fails to detect hard OOD samples, and add them in the updated manuscript (**line 525 and Fig.6**). For example, in the first image, previous global-based method fails to detect the outlier as it is similar to ID category ocean liner on the whole, with only subtle difference mechanical devices on the deck indicating that it is actually an icebreaker (also demonstrated by the iceberg next to it). The same phenomenon are also observed other examples (the shape of the sunflower center in the second image and the calyx of the apple in the third image). From the examples, it is vividly illustrated in the example that global-based method focuses on overall representations and fails in hard outliers with subtle difference in regions. By contrast, our method enhances local information and successfully solves the problem by outlier knowledge, qualitatively showcasing the superiority and effectiveness of our method.

---

> ### Author Response · Authors · 2024-11-22
> **Author Response to Reviewer NiJR for Q4-Q5**
>
> #### **Q4: Influence of the two hyperparameters.**
> A4: In Table.9, we aim to explain that the proposed method is not sensitive to the hyperparameter in a wide range, **at least in the entire ablation process of our practical experiment** (in our experiment 1-10 for $\lambda_{neg}$ and 0.1-1 for $\lambda_{reg}$). Empirically, we find that the ratio of the two corresponding coefficients should be within a wide range (approximately 2-10), and do not carefully select the hyperparameters. We use the two coefficients just for rigorous description of the entire training procedure in Eqn.8 and we highlight that our method still gets robust results without any coefficient parameter ( $\lambda_{neg}=1$, $\lambda_{reg}=1$ in table below) and outperforms previous methods like LoCoOp. We also carry out ablation study about $\lambda_{neg}$ to demonstrate the effectiveness of different modules.
>
> |method|iNaturalist||SUN||Places||Texture||Average||
> |-|-|-|-|-|-|-|-|-|-|-|
> ||FPR|AUORC|FPR|AUORC|FPR|AUORC|FPR|AUORC|FPR|AUORC|
> |Ours (main paper)|9.65| 97.87| 20.40 |95.57 |29.39| 92.67 |51.20 |88.00 |**27.66** |**93.53**|
> |Ours (w/o $\lambda_{neg}$)| 9.84 |97.88| 24.37| 94.97| 32.84 |91.94| 50.02 |88.32 |**29.27**| **93.27**|
> |Ours ($\lambda_{neg}=1$, $\lambda_{reg}=1$)|12.24|97.53|25.11|95.23|31.24|92.46|50.03|88.46|**29.65**|**93.42**|
>
>
> #### **Q5: Nonlinear change in FPR95.**
> A5: the attribute of nonlinear characteristics in few-shot learning has been widely observed in the fields of OOD detection[1]. For example, **similar trend is observed** in ID-like[2] (1-shot and 4-shot in Places shown in table below) and LoCoOp[1] (4 shot and 16 shot in SUN and Places shown in Table.1). The reason for the phenomenon could be: **1)** the effectiveness of few-shot learning that effectively learns knowledge of downstream tasks especially in the very few samples;  **2)** Moreover, for few-shot OOD detection, more samples may not necessarily bring better performance, and incorporating more samples may confuse the discrimination, which should **be the characteristic of the two datasets**. In Table.8, the reason for the nonlinear change can partly be attributed to that our local-based model learns characteristics of OOD detection with efficiency and gets the best discrimination performance between ID and OOD in FPR95, and more samples just confuse the process.
>
>
> |method|iNaturalist||SUN||Places||Texture||
> |-|-|-|-|-|-|-|-|-|
> ||FPR|AUORC|FPR|AUORC|FPR|AUORC|FPR|AUORC|FPR|AUORC
> |ID-like (1 shot)|14.57 |97.35|44.02 |91.08| 41.74 |91.15 |26.77| 94.38|
> |ID-like (4 shot)| 8.98 |98.19|  42.03 |91.64| 44.00| 90.57 |25.27 |94.32| 26.08| 94.36|
>
> [1] Few-Shot Out-of-Distribution Detection via Prompt Learning. NeurIPS 2023.
>
> [2] ID-like Prompt Learning for Few-Shot Out-of-Distribution Detection. CVPR 2024.

---

> ### Author Response · Authors · 2024-11-25
> **Looking forward to your feedback**
>
> Dear reviewer NiJR:
>
> We respectfully appreciate again for your insightful and thoughtful comments! As the suggestions from the reviewer, we give thorough explanation and update the manuscripts accordingly.
>
> As it is approaching the end of the discussion period (November 26 at 11:59 pm AoE) and we do not receive feedback, we sincerely hope you could look through our response and have a further comment at your convenience if you have any questions about the paper. We will do our best to address the issues of the reviewer.
>
> Best wishes,
>
> Submission 6694 Authors.

---

> > ### Comment · Reviewer_NiJR · 2024-11-25
> >
> > I would like to thank the authors for their detailed responses. The responses addressed some of my concerns, so I would like to retain my original rating. I still have the following concerns.
> >
> > **W2**. In Table 2 newly presented in W2, the differences between Ours w/o LNP and Ours w/ LNP are all less than 1.0 point. For example, given that LoCoOp shows an average improvement of more than 2.0 points over its competitors based on its core technical idea, I feel that the improvements made by the proposed method are rather marginal. In addition, I would say Fig. 4 should be replaced by a table to quantify the differences between the methods.
> >
> > **W3**. I would like to thank the authors for providing new visualization results. However, at least two of the examples are due to clear misclassification of objects. I rather feel that it would be preferable if examples were those that are correctly identified even when viewed globally, but cannot be determined to be OOD unless viewed locally.
> >
> > **W4**. The authors' response is understandable. However, if so it should be clearly stated in the text that "at least in the entire ablation process of our practical experiment." Actually the authors' response does not really address my point. The key of the proposed method is in the loss term. If the weight $\lambda_{\text{neg}}$ for the loss term does not change the final performance, it could negate the impact of the proposed method. To dispel this contradiction, the analysis should be performed for a sufficiently wide range of $\lambda_{\text{neg}}$ for which the contribution to performance becomes clear.
> >
> > Last but not least, reading the fellow reviewer's comments, I agree with Reviewer dbE1's comment that the novelty of the proposed method is somewhat diminished because it is similar to the existing methods such as ID-like that use randomly cropped regions to supervise positive and negative prompt learning.

---

> > > ### Author Response · Authors · 2024-11-26
> > > **Additional Response to Reviewer NiJR for Q2-Q3**
> > >
> > > We are grateful to the reviewer for taking the time to view the rebuttal and having discussion with us. We sincerely thank the valuable suggestions from the reviewer and we answer remaining questions in detail sequentially.
> > >
> > > A2: We would like to make a clarification that ours with LNP is also an OOD metric proposed by us. The two metrics both integrate global and local information. Concretely, the local prompts are also trained by our proposed method and the difference lies in the utilization of local negative prompts. Specific definitions are shown below.
> > >
> > > $S_{\mathrm{R-MCM} w LNP}(\boldsymbol{x})$=$S_{\mathrm{MCM}}(\boldsymbol{x})$+ $\mathcal{T_k^{\mathrm{mean}}}(\mathrm{exp}(\mathrm{sim}(z_h^l, t_i)/ T)/(\sum_{j=1}^C{\mathrm{ exp}(\mathrm{sim}(z_h^l, t_j) / T)}+\sum_{j=1}^{N_{\mathrm{neg}}}{\mathrm{exp}(\mathrm{sim}(z_h^l, \hat{t}_j) / T)}))$
> > >
> > > and
> > > $S_{\mathrm{R-MCM} w/o LNP}(\boldsymbol{x})$ =$S_{\mathrm{MCM}}(\boldsymbol{x})$+ $\mathcal{T_k^{\mathrm{mean}}}(\mathrm{exp}(\mathrm{sim}(z_h^\mathrm{l}, t_i)/ T)/\sum_{j=1}^C{\mathrm{ exp}(\mathrm{sim}(z_h^l, t_j) / T)})$
> > >
> > > We hold the view that utilization of local prompts (enhance local information), local negative prompts (provide outlier knowledge) and both local-based metrics, i.e., ours w/o LNP and ours w LNP are all our contributions. We give a detailed comparison of the proposed method with LoCoOp in a progressive manner. If the reviewer wants to figure out the influence of the proposed method, the comparison should be within the first two lines below:
> > >
> > >
> > > |method|iNaturalist||SUN||Places||Texture||Average||$\Delta$|
> > > |-|-|-|-|-|-|-|-|-|-|-|-|
> > > ||FPR|AUORC|FPR|AUORC|FPR|AUORC|FPR|AUORC|FPR|AUORC|
> > > |LoCoOp_{GL}|21.67| 95.69 |22.98 |95.07 |31.41| 92.10 |49.79| 87.85 |31.46 |92.68||
> > > |Ours_{GL}|9.69|97.80|26.27|94.22|34.78|91.33|36.63|91.65|26.84| 93.75|**4.62%**|
> > > |Ours_{R-MCM}(w/o LNP)|8.91|97.88|23.78|94.98|32.94|92.02|35.76|91.79|25.34|94.16|**6.12%**|
> > > |Ours_{R-MCM}(w LNP)|8.63|98.07|23.23|95.12|31.74|92.42|34.50|92.29|**24.52**|**94.48**|**6.94%**|
> > >
> > > The first two lines indeed showcase the benefits from our local prompts with enhanced local information **(4.62% on FPR95)**. Once more, equipped with two kinds of OOD metrics proposed by our paper, we are able to further boost the performance **(2.32% on FPR95)**. Even without local negative prompts (serve as an ablation), the local prompts are already capable of improving OOD performance with fine local information. Therefore, we would say not only the improvements from utilizing LNP are the core contribution but also the local prompt with enhanced local information. We sincerely hope that the reviewer could have a better understanding of our core contribution from the comparison table above.
> > >
> > >
> > > A3: We would like to make a clarification of the new visualizaition results. These samples are globally classified into an ID category with high confidence and therefore fail to be an OOD sample by previous global-based methods.
> > >
> > > We can not fully understand the meaning of the reviewer. What is the meaning of the reviewer by mentioning "samples that are correctly identified, but cannot be determined to be OOD"? Does it mean 1) the samples are correctly classified into ID-categories globally or 2) correctly classified between ID and OOD? If the reviewer means the latter, there would be no doubts for "but cannot be determined to be OOD", so we assume the reviewer means the former. We would explain that OOD samples can not be classified correctly as there are no OOD categories in advance. **The only difference is whether it is assigned to an ID category with high confidence.** If it is, then it would be just the samples in new visualization results, i.e., global-based methods fail in some cases that are globally similar to certain ID categories. By contrast, our method successfully discriminates these circumstances with local information and treats it as OOD. Once one certain example is classified correctly, it means the sample is indeed an ID sample, and **it is determined not to be OOD**. We select three cases and all of them are outlier samples. They all **can not be classified correctly as no ID categories match their categories**. In other words, none of them can be correctly identified no matter globally or locally. However, when globally identified, they are incorrectly assigned to ID classes that are overall similar to, and no longer be considered as OOD. By contrast, our method with local enhancement successfully addresses the issue.
> > >
> > > We are uncertain if we get the meaning of the reviewer. If our understanding differs from that of the reviewer, please feel free to let us know and we will make an instant explanation accordingly.

---

> > > ### Author Response · Authors · 2024-11-26
> > > **Additional Response to Reviewer NiJR for Q4**
> > >
> > > A4: We thank the reviewer's kind suggestion and make a clarification of the range in the updated manuscript. As for the influence of the loss term, we follow the suggestion from the reviewer to conduct a sufficiently wide range of $\lambda_{neg}$ (**where some extreme cases are rarely considered in practical applications because they are too unbalanced.**)
> > >
> > > |method|iNaturalist||SUN||Places||Texture||
> > > |-|-|-|-|-|-|-|-|-|
> > > |$\lambda_{neg}$|FPR|AUORC|FPR|AUORC|FPR|AUORC|FPR|AUORC|FPR|AUORC
> > > |0.1|17.64|95.97|27.18|94.76|34.45|91.41|48.82|86.89|
> > > |1|9.92| 97.80| 25.04| 94.86| 32.83| 92.02| 48.90 |88.79
> > > |10|10.19| 97.83| 23.46| 95.26| 31.59| 92.31| 49.57| 88.81
> > > |50|34.09|92.06|42.34|88.63|50.88|83.52|62.47|85.70|47.44|
> > >
> > >
> > > It can be seen that the coefficient indeed works, with performance drop under extreme cases ($\lambda_{neg}=0.1$ and $50$). For $\lambda_{neg}=50$, the loss for local prompts is neglected, resulting in severe performance drop. For $\lambda_{neg}=0.1$, the local negative prompts are over-regularized by diversity constraints. The effectiveness of the proposed loss can be further verified accordingly. We thank the reviewer for the remind and revise the statement in the corresponding paragraph.
> > >
> > >
> > > As for the comparison with ID-like, we provide comprehensive analysis from the perspective of motivation and empirical experiment just as stated in response to reviewer dbE1. We believe that the core contribution of our method **lies in the utilization of local information with outlier knowledge**. Despite employing random crop, ID-like still concentrates on OOD detection from the perspective of global information. By contrast, all the module and regional loss function of our method are designed to meet the demand for incorporating local information. In fact, the **first visualization in A3 is from misclassification of ID-like in its paper**. It strongly demonstrates the effectiveness of our method in discriminating OOD samples with subtle differences in certain regions which global-based methods cannot. Empirically, we show that our method achieves substantial **3.78% improvements regarding FPR**. Moreover, the unique extensibility of our method can further **boost the performance with the aid of any global-based method**, and again achieves **impressive 6.41% improvements on FPR95**.
> > >
> > > Last but not least, please allow us to explain that improving OOD performance is challenging, and we demonstrate empirically that promoting 2% of FPR95 performance approximately equals to promotion from 4-shot to 16-shot. Despite the challenges ahead, we still **get state-of-the-art results after comprehensive comparison with all existing methods**, strongly demonstrating the effectiveness of the entire pipeline of the proposed method. Moreover, our unique implementation from the perspective of local prompt enhancement equips us with the ability to further **integrate well-trained global prompts from all existing methods**, which means we are always able to enhance OOD detection in a way like "standing on the shoulders of giants". We believe that the local enhancement could serve as a direction that could be well explored to make a breakthrough in OOD detection.
> > >
> > >
> > > We are sincerely grateful that the reviewer takes the time to view the rebuttal, and we respond to each of the remaining issues respectively. We hope the additional explanation and experimental results can address the problems. If the reviewer still has concerns, let us know in detail and we will make a further explanation.

---

> > > > ### Comment · Reviewer_NiJR · 2024-12-02
> > > >
> > > > Thanks to the authors for further responses, and sorry for the last minute reply. Some of my questions have been successfully clarified.
> > > >
> > > > **W3**. In the first example in Fig. 6, the global classification result (“Ocean”) correctly reflects the content of the image, and we can never say that the result is wrong. Given that the groundtruth of the image is "icebreaker," which can be determined by focusing on a local region of the image, it is reasonable for the proposed method to classify the image as "OOD." On the other hand, in the second example, the global classification result is "Daisy," which is a clear misclassification. Due to the differences, I found the second (or third) example less convincing.
> > > >
> > > > Regarding the novelty of the proposed method to ID-Like, I have checked the responses and the comparative results provided to Reviewer dbE1. I think I understood the author's points. However, I still think that there are significant conceptual and technical overlaps between the two methods, which undermines the novelty of the proposed method.

---

> ### Author Response · Authors · 2024-12-02
>
> Dear reviewer NiJR:
>
> We would like to express our gratitude for your kind response and make further explanation about your concerns. We will explain them sequentially.
>
> With respect to W3,  we would like to emphasize that correctly classification with regional difference is just **one of the reasons that cause wrong OOD detection results, and other type of wrong OOD detection should not be ignored**. We understand the meaning of the reviewer that the OOD detection fails mainly because ocean liner and icebreaker both belong to "ship" in some way (kind reminder that ocean liner is acutally a wrong result because the icebreaker cannot serve as a passenger ship). In other words, previous method fails in **this kind of outliers** because they share similar global meaning.
>
> However, **it is not the only way that previous global-based OOD detection method fails**. It also indeed comes from the misclassification where ID and OOD are also with subtle differences in certain regions. This kind of wrong detection is also challenging and by no means easier than the first type. For example, in the third example, the ID (Rose hip) is almost the same as OOD (Apple) and is **hard to discriminate even from the perspective of local information**. It should not be dismissed that the difficulty in discriminating them is as hard as the first type of example and that ID and OOD samples **do not necessarily belong to similar parent class**. Here we just want to cover more types of sources that cause wrong OOD detection results. We thank the reviewer's suggestion and will provide more visualizations in the updated manuscripts.
>
> With the help of visualizations, we also would like to further illustrate the uniqueness of our method. The first example that the reviewer agrees to be persuasive **just comes from the visualization of the original paper** (we are unable to provide an external link but we sincerely wish the reviewer to look through it at the third example of Fig.7 on page 8). We believe it firmly demonstrates that the best global-based OOD detection method can only concentrate on outlier samples with overall differences and fails in hard samples that only have subtle differences in certain regions, whichi is the core issue that we aim to address.
>
> Therefore, we would like to express that the **core motivation and contribution of our method lies in the utilization of refine local information**, and the training pipeline and design of loss function to achieve this goal, which is not explored before. The motivation and starting point of the two method is totally different, and the so-called "negative samples" is just the way to generate pseudo samples. Empirically, we adequately demonstrate that our local-based method achieves competitive results and further gains substantial improvements with no doubts. We sincerely hope that our explanation could give the reviewer a better understanding that **1) outlier information from local perspective and 2) extensibility to integrating global prompts to get further improvements** is the core contribution of our paper. We will add more explanation of the comparison with the method in the updated manuscript.

---

### Official Review · Reviewer_dbE1 · 2024-11-03

**Soundness:** 2
**Presentation:** 3
**Contribution:** 2
**Rating:** 6
**Confidence:** 4

**Summary:**

The proposed work focuses on enhancing the model's ability to detect local features by training local feature prompts using localized image information. Furthermore, based on these local feature prompts, this paper introduce a new OOD score that integrates with MCM Score.

**Strengths:**

1.	Local image information is utilized to train local feature prompts, enhancing the model's ability to detect local features.

2.	Based on local feature prompts, an OOD metric combining MCM has been proposed, along with a new ID classification metric.

**Weaknesses:**

1.	In the latest works related to OOD detection, there are several studies similar to the method in this paper, such as "ID-like (ID-like Prompt Learning for Few-Shot Out-of-Distribution Detection)" and "NegLabel (NEGATIVE LABEL GUIDED OOD DETECTION WITH PRETRAINED VISION-LANGUAGE MODELS)." The structure of this paper is similar to that of ID-like, as it also employs random crop combined with CLIP to construct ID and OOD data, and utilizes positive prompts and negative prompts. However, the paper lacks a comparison and analysis of strengths and weaknesses with the ID-like paper. It is recommended that the authors add relevant discussions to enhance the depth and breadth of the paper.

2.	The OOD Score strategy proposed in this paper presents results for SMCM and SR-MCM but lacks results that only use the Regional OOD score and corresponding ablation experiments. This makes it difficult to effectively demonstrate that the SR-MCM outperforms the strategy that solely relies on the Regional OOD score. It is suggested that the authors include this part of the experiment to strengthen the paper's persuasiveness and the comparability of the results.

**Questions:**

Please refer to the weakness.

---

> ### Author Response · Authors · 2024-11-22
> **Autrhor Response to Reviewer dbE1 for Q1(1/2)**
>
> We thank the reviewer for the constructive comments and respond to them appropriately as follows.
>
> #### **Q1: Analysis of related works.**
> A1: As the reviewer suggests, we compare with related work ID-like prompt and NegPrompt and respectively show the detailed analysis.
>
> **1) ID-like**
>
> We conduct thorough comparison with ID-like[1]. First, **from the perspective of motivation**, our method differs from ID-like in that ID-like still focuses on global prompts optimization, with random crop generating ID-like samples. As it treats the whole image overall, it can not solve the problem of hard OOD samples with subtle differences in certain regions as described in line52-53. By contrast, our method optimizes local prompts and can well integrate with ID-like to further enhance the performance. **Empirically**, we compare OOD detection performance and showcase the results in the table below.
>
> |method|iNaturalist||SUN||Places||Texture||Average||
> |-|-|-|-|-|-|-|-|-|-|-|
> ||FPR|AUORC|FPR|AUORC|FPR|AUORC|FPR|AUORC|FPR|AUORC
> |ID-like (global prompt optimized)|8.98 |98.19 |42.03 |91.64|44.00 |90.57 |25.27| 94.32| 30.07| 93.68
> |Ours (local prompt optimized)|12.81|97.29|19.34|95.85|27.53|92.97|45.51|89.99|**26.29** |**94.03**|
> |Ours+ID-like| 8.19|98.58| 19.77| 95.01| 28.92| 93.01|37.76| 91.57|**23.66**|**94.54**|
>
> It can be concluded that:
> **1)** our model achieves better average performance against ID-like, demonstrating the utility of our local information enhancement strategy; **2)** ID-like obviously performs poorly in SUN and Places, which mainly consist of scenery with diverse scenarios; **3)** integrating the global prompt into our model further enhances the performance as shown in the last row. We attribute it to the advantage that it brings well-trained global prompts that better tailor for OOD detection rather than the hand-crafted template "a photo of {class}". It is helpful to our method in distinguishing outlier samples that have overall-dominant features, further showcasing unique advantage of local prompts and the potential of local prompt optimization as an orthogonal direction for OOD detection.
>
> We additionally showcase the examples that are not detected by previous global based method in Fig.6. By contrast, the subtle differences can be discriminated by our method, which further demonstrates the effectiveness of our method.

---

> ### Author Response · Authors · 2024-11-22
> **Author Response to Reviewer dbE1 for Q1(2/2)**
>
> **2) NegLabel**
>
> Neglabel[2] designs a novel scheme for the OOD score with negative labels. It leverages **real outlier information** with negative labels from extensive corpus databases, which we discuss in line38-41. This kind of knowledge helps to a great extent pointed out by OE[3] and is inconsistent with real-world application, where negative categories are infinite. As we have **no access to any true outlier information**, the comparison is unfair. Moreover, while all of the mentioned works mainly focus on negative prompts, we highlight that we concentrate on the connection between global and local, and negative prompt is a small part of our method. By contrast, the advantage of our method is that it incorporates local information with pseudo local outliers and finds a novel way to take local information into both training and OOD score calculation. As the reviewer suggests, we provide comparison and analysis shown below:
>
> |method|iNaturalist||SUN||Places||Texture||Average||
> |-|-|-|-|-|-|-|-|-|-|-
> ||FPR|AUORC|FPR|AUORC|FPR|AUORC|FPR|AUORC|FPR|AUORC
> |NegLabel (w outlier)|1.91|99.49|20.53|95.49|35.59|91.64|43.56|90.22|25.40|94.21|
> |Ours (w/o outlier)|8.63|98.07|23.23|95.12|31.74|92.42|34.50|92.29|**24.52**|**94.48**|
>
> We observe that **1)** although without outlier, our model achieves better average performance against NegLabel, demonstrating the effectiveness of our local information refinement strategy; **2)** NegLabel has an obvious overfitting in iNaturalist and SUN, which mainly consist of natural scenery. These datasets share different data distributions with images like Texture, which are detailed texture images. Consequently, our method achieves better balance between different kinds of OOD datasets, which strongly validates the effectiveness of incorporating local information and strengthens its application in diverse and infinite read-world scenarios.
>
> We include these two relevant works in the revised version (line 426-447). We hope the comparison and analysis would help reviewer have a better understanding of the unique contributions and advancements of the proposed method.
>
>
> We would like to emphasize that the core is to better utilize local outlier information to strengthen the ability of OOD detection, which has not been explored before, and some kind of **negative prompts** are just the way to achieve it. Moreover, our method **can well adapt to all global prompt-based methods (see the experiments in Table.1 and above)** and integrating with them achieves better performance, which strongly demonstrates the potential and generalization ability. Despite the similarity on the surface, the starting point is essentially different. We hope the analysis above can help the reviewer have a better understanding of the strengths and unique contributions of our paper. We are grateful that the reviewer points out the related work and we update in the revised manuscripts.

---

> ### Author Response · Authors · 2024-11-22
> **Author Response to Reviewer dbE1 for Q2**
>
> #### **Q2: Results using Regional OOD score.**
>
> A2: Following the suggestion of the reviewer, we additionally conduct experiment of OOD score solely dependent on Regional OOD score. Results are shown in table below.
>
> |method|iNaturalist||SUN||Places||Texture||
> |-|-|-|-|-|-|-|-|-|
> ||FPR|AUORC|FPR|AUORC|FPR|AUORC|FPR|AUORC|FPR|AUORC
> |MCM|30.91| 94.61| 37.59| 92.57| 44.69| 89.77| 57.77| 86.11|
> |Ours (Regional)|39.17|88.04| 67.06|82.47|60.13|79.61|80.96|67.35|
> |Ours (Global)|27.70|94.66|31.30|93.68|39.50|90.86|38.61|91.79|
> |Ours (Global+Regional)| **9.65**| **97.87** |**20.40**|**95.57**|**29.39**| **92.67**| **51.20**| **88.00**|
>
> From the table, it can be concluded that **1)** Regional OOD score can not achieve satisfying results in our experiment and must rely on global score; **2)** the proposed R-MCM achieves the best results with respect to all metrics. This is in line with analysis that global prompt/score provides **global and fundamental** discrimination between ID and normal OOD samples (Sec.4.1) and local prompts/score provides **fine** discrimination results between ID and hard OOD samples (Sec.4.2). Consequently, it demonstrates that our proposed score integrating gliobal and local information outperforms the strategy that solely relies on the Regional OOD score. We kindly highlight that we aim to propose a novel approach to enhance OOD detection performance from the perspective of enhancing local information but not to demonstrate that local prompts are better/more important than global prompts and can replace their position.
>
>
> We would appreciate it if our explanation could address the review's concerns. If you have any other questions about the method or the experiment, reply to us and we will try our best to address your confusion.
>
>
> [1] ID-like Prompt Learning for Few-Shot Out-of-Distribution Detection. CVPR 2024.
>
> [2] Negative Label Guided OOD Detection with Pretrained Vision-Language Models. ICLR 2024.
>
> [3] Deep Anomaly Detection with Outlier Exposure. ICLR 2019.
>
> [4] Zero-Shot In-Distribution Detection in Multi-Object Settings Using Vision-Language Foundation Models.

---

> ### Author Response · Authors · 2024-11-25
> **Looking forward to your feedback**
>
> Dear reviewer dbE1:
>
> We respectfully appreciate again for your insightful and thoughtful comments! As the suggestions from the reviewer, we give thorough explanation and update the manuscripts accordingly.
>
> As it is approaching the end of the discussion period (November 26 at 11:59 pm AoE) and we do not receive feedback, we sincerely hope you could look through our response and have a further comment at your convenience if you have any questions about the paper. We will do our best to address the issues of the reviewer.
>
>
> Best wishes,
>
> Submission 6694 Authors.

---

> ### Author Response · Authors · 2024-11-30
>
> Dear reviewer dbE1:
>
> We sincerely appreciate again for taking the time to provide valuable suggestions for our work! As the suggestions from the reviewer, we give thorough comparison with the mentioned papers to demonstrate the strength of the paper both theoretically and empirically and and update the manuscripts accordingly (Sec.5.1). We additionally carry out experiments to validate the effectiveness of the proposed OOD metric (shown in the first-round discussion).
>
> As it is approaching the end of the discussion period and we do not receive feedback, we are not sure if our responses address the concerns from the reviewer. We sincerely hope you could look through our response. We would be grateful if you could have a further comment at your convenience. If you have any questions about the paper, We will do our best to address them.
>
> Best wishes,
>
> Submission 6694 Authors

---

> ### Author Response · Authors · 2024-12-02
>
> Dear Reviewer dbE1:
>
> Thank you for taking the time to review our paper and propose constructive suggestion.
> Could you please kindly review our updated manuscript or previous comments? We have provided detailed responses to each question raised by the reviewer and we believe we have addressed all the concerns in our updated paper and previous responses. Thus, we would appreciate it if the reviewer could view the rebuttal and reevaluate the score for our paper. If the reviewer has more questions, we will try our best to address the problems.
>
> Best wishes,
>
> Submission 6694 Authors

---

### Author Response · Authors · 2024-11-22
**General response to all reviewers**

We are grateful to all reviewers for taking the time to review our work and highly recognize our contributions: "idea is novel, shed light on an interesting perspective of local features in OOD detection" (**dbE1**,**NiJR**,**GNZR**), "the effectiveness of the method is extensively demonstrated" (**KX2t**,**GNZR**), "the paper is well-organized and easy to follow" (**NiJR**,**KX2t**). We update the manuscript and highlight all the changes from the feedback of the reviewers with blue color. We will respond to each reviewer respectively and try our best to address the concerns of the reviewers.

---

### Meta-Review · Area_Chair_sQtt · 2024-12-25

**Metareview:**

This work aims to enhance few-shot tuning for OOD detection by complementing global prompts with regional enhancements through local prompts. To achieve this, the authors introduce two key modules: (1) negative augmentation to leverage local outlier knowledge, and (2) local prompt-enhanced region regularization to effectively capture local information. The extensive results have demonstrated the effectiveness of the proposed approach. Several reviewers acknowledged the novelty of the proposed method (dbE1, NiJR, GNZR), its effectiveness in improving OOD detection (KX2t, GNZR), and the clarity of the paper’s organization, which facilitates understanding. However, reviewers raised a number of concerns, including the need for additional comparisons with similar methods (e.g., ID-like, NegLabel) (dbE1, KX2t), further explanations and justifications regarding the proposed approach (KX2t, GNZR), and more ablation studies (e.g., w/ or w/o LNP) (dbE1, NiJR, GNZR). Additional concerns were noted regarding hyperparameter influence analysis (i.e., $\lambda_{neg}$ and $\lambda_{reg}$) (NiJR), limited performance gains (dbE1, NiJR), and performance degradation in few-shot scenarios (e.g., declining performance from 4-shot to 8-shot or higher) (NiJR, GNZR). The authors provided detailed responses and effectively addressed most of these concerns. As a result, all reviewers ultimately gave positive ratings, leading to an average score of 6.0. We decide to accept the paper. Additionally, the authors are encouraged to further refine the manuscript in accordance with the reviewers' suggestions to strengthen the final version.

**Additional Comments On Reviewer Discussion:**

During the discussions, the authors effectively addressed the concerns raised by reviewers NiJR and KX2t, leading both to increase their scores. Reviewer GNZR maintained some reservations regarding performance degradation with additional shots and the limited performance gains observed. The authors provided follow-up responses to further clarify and mitigate these concerns. Reviewer dbE1 did not participate in the discussion phase. Overall, all reviewers ultimately gave positive ratings, and the authors’ detailed and thorough responses successfully addressed the majority of the feedback. Based on these factors, I agree that the proposed method demonstrates sufficient merit for publication.

---

### Decision · Program_Chairs · 2025-01-22

Accept (Poster)